# Design, Synthesis and Biological Evaluation of Novel Pleuromutilin Derivatives Containing 6-Chloro-1-R-1*H*-pyrazolo[3,4-*d*]pyrimidine-4-amino Side Chain

**DOI:** 10.3390/molecules28093975

**Published:** 2023-05-08

**Authors:** Jun Wang, Yu-Han Hu, Ke-Xin Zhou, Wei Wang, Fei Li, Ke Li, Guang-Yu Zhang, You-Zhi Tang

**Affiliations:** 1Guangdong Provincial Key Laboratory of Veterinary Pharmaceutics Development and Safety Evaluation, College of Veterinary Medicine, South China Agricultural University, No. 483 Wushan Road, Guangzhou 510642, China; wangjun2261@126.com (J.W.);; 2Guangdong Laboratory for Lingnan Modern Agriculture, Guangzhou 510642, China

**Keywords:** antibacterial activity, MRSA, pleuromutilin, 1*H*-pyrazolo[3,4-*d*]pyrimidine

## Abstract

Two series of pleuromutilin derivatives were designed and synthesized as inhibitors against *Staphylococcus aureus* (*S. aureus*). 6-chloro-4-amino-1-R-1*H*-pyrazolo[3,4*-d*]pyrimidine or 4-(6-chloro-1-R-1*H*-pyrazolo[3,4*-d*]pyrimidine-4-yl)amino-phenylthiol were connected to pleuromutilin. A diverse array of substituents was introduced at the N-1 position of the pyrazole ring. The in vitro antibacterial activities of these semisynthetic derivatives were evaluated against two standard strains, Methicillin-resistant *Staphylococcus aureus* (MRSA) ATCC 43300, *Staphylococcus aureus* (*S. aureus*), ATCC 29213 and two clinical *S. aureus* strains (144, AD3) using the broth dilution method. Compounds **12c**, **19c** and **22c** (MIC = 0.25 μg/mL) manifested good in vitro antibacterial ability against MRSA which was similar to that of tiamulin (MIC = 0.5 μg/mL). Among them, compound **22c** killed MRSA in a time-dependent manner and performed faster bactericidal kinetics than tiamulin in time–kill curves. In addition, compound **22c** exhibited longer PAE than tiamulin, and showed no significant inhibition on the cell viability of RAW 264.7, Caco-2 and 16-HBE cells at high doses (≤8 μg/mL). The neutropenic murine thigh infection model study revealed that compound **22c** displayed more effective in vivo bactericidal activity than tiamulin in reducing MRSA load. The molecular docking studies indicated that compound **22c** was successfully localized inside the binding pocket of 50S ribosomal, and four hydrogen bonds played important roles in the binding of them.

## 1. Introduction

The development of bacterial resistance to available antibiotics has been growing at a high speed, becoming one of the greatest public health problems. Methicillin-resistant *Staphylococcus aureus* (MRSA) has become one of the most important pathogens, which was first identified in the 1960s [1,2]. MRSA can cause a variety of diseases, ranging from skin infections to life-threatening invasive infections, such as pneumonia, endocarditis and sepsis [3]. MRSA infections are associated with high mortality [4]. There were about 120,000 bloodstream infections caused by MRSA in the USA in 2017, and nearly 20,000 of those affected lost their lives [5]. MRSA has been reported to be resistant to many clinical antibiotics, including linezolid, vancomycin, penicillin and all commonly prescribed beta-lactam antibiotics [6]. With the prevalence of MRSA, new antimicrobial agents are urgently needed to address its increasingly serious drug resistance.

Pleuromutilin (**1**, Figure 1), containing a tricyclic core of five-, six- and eight-membered rings, was first isolated from two basidiomycete species Pleurotus mutilus and Pleurotus passeckerianus in 1951 [7]. Pleuromutilin has been proven to be effective against Gram-positive pathogens [7]. It has been identified that pleuromutilin inhibits bacterial protein synthesis through its interaction with the 50S ribosomes [8,9]. This distinctive antibacterial mechanism endowed pleuromutilin with great potential to deal with drug-resistant bacterial infections, and encouraged researchers to develop novel pleuromutilin derivatives as effective antibacterial agents [10]. It was found that the modification of the C-14 acyloxy side chain significantly affected the antibacterial activity of pleuromutilin [11]. Based on this modification strategy, four of them have reached the market: tiamulin (**2**, Figure 1), valnemulin (**3**, Figure 1), retapamulin (**4**, Figure 1) and lefamulin (**5**, Figure 1) [12,13]. Tiamulin and valnemulin have been used to treat economically important infections in swine and poultry, which were approved in 1979 and 1999, respectively [14,15]. Success in veterinary antibiotics has encouraged researchers to focus on the development of new pleuromutilin antibiotics for human use. Retapamulin is the first pleuromutilin antibiotic for human skin infections caused by *S. aureus* [16]. In 2019, lefamulin was approved as an oral and intravenous pleuromutilin antibiotic for the treatment of community-acquired bacterial pneumonia in humans [17].

Our lab is devoted to the development of pleuromutilin derivatives containing nitrogenous bases in the C14 side chain. Previous work has led to quite a few target products with powerful anti-MRSA activity. Fused N-heterocycles are important heterocyclic compounds with a wide spectrum of bioactivities, such as anti-cancer (ibrutinib, **6**, Figure 1) and anti-bacterial (formycin A, **7**, Figure 1) [18,19,20,21]. Among them, pyrazolo[3,4-*d*]pyrimidine (**8**, Figure 1) derivatives possess antibacterial pharmacological activity [22,23]. These motivated us to develop pleuromutilin derivatives contenting pyrazolo[3,4-*d*]pyrimidine moiety (Figure 1). Additionally, the introduction of 4-aminothiophenol has been reported to influence the antimicrobial activity of pleuromutilin [24]. Thus, another synthetic strategy is based upon attaching 4-aminothiophenol as a linker arm to the pleuromutilin and attaching the other end to a pyrazolo[3,4-*d*]pyrimidine moiety via an amino group (Figure 2). In this work, 34 novel pleuromutilin derivatives were designed and synthesized via 2 diverse strategies, and we preliminarily evaluated their in vitro and in vivo antibacterial activity. The interactions between the derivatives and 50S ribosomes were imitated by molecular docking.

## 2. Results and Discussion

### 2.1. Chemistry

The general synthesis routes of target products were illustrated in Figure 1 and Figure 2. Urea (compound **9**) was used as starting material. Barbituric acid (compound **10**) was obtained via the condensation reaction of urea and diethyl malonate [25]. Trichloropyrimidine (compound **11**) was prepared using the Vilsmeier–Haack reaction [26]. Pyrazolo[3,4-*d*]pyrimidines (compounds **12a**~**28a**) were acquired via the cyclization reaction of 2,4,6-trichloropyrimidine-5-carbaldehyde (compound **11**) with different substituted hydrazines [27,28]. The structure of compounds **12a**~**28a** is disclosed in Appendix A.

As shown in Figure 1, the reaction of pleuromutilin (compound **1**) with *p*-toluenesulfonyl chloride afforded compound **29** under alkaline condition [29]. Substitution of the *p*-toluenesulfonyl group with sodium azide provided compound **30**, which subsequently underwent Staudinger reduction with triphenylphosphine to produce compound **31** [30]. Target compounds **12b**~**28b** were synthesized via the nucleophilic substitution reaction of compounds **12a**~**28a** and compound **31**. As shown in Figure 2, 22-(4-Amino-phenylsulfanyl)deoxypleuromutilin (compound **32**) was acquired in reference to our previous work [24]. Target compounds **12c**~**28c** were synthesized by a method similar used for compounds **12b**~**28b** [28].

The pleuromutilin derivatives were purified by silica gel column chromatography and characterized by ^1^H NMR, ^13^C NMR and high-resolution mass spectral (HR-MS) analysis. The results confirmed that the synthesis of compounds was consistent with the expected structure. All the spectra of synthesized pleuromutilin derivatives are supplied in the Appendix A.

### 2.2. In Vitro Antibacterial Activity

The in vitro antimicrobial activity of the pleuromutilin derivatives was assessed against MRSA ATCC 43300, *S. aureus* ATCC 29213, *S. aureus* 144 and *S. aureus* AD3 according to the Clinical and Laboratory Standards Institute (CLSI) [31]. Tiamulin was used as a positive control drug. The results were summarized as the minimum inhibitory concentration (MIC) and the minimum bactericidal concentration (MBC) in Table 1 and Table 2. In addition, MIC results for the pleuromutilin derivatives against four Gram-negative bacteria are supplied in Appendix A.

As illustrated in Table 1 and Table 2, the antimicrobial activity of most compounds was absent. Compounds **12c**, **19c** and **22c** (MIC = 0.25 μg/mL) exhibited better anti-bacterial effects than tiamulin (MIC = 0.5 μg/mL). Meanwhile, the preliminary structure-activity relationships (SARs) were also studied. Two schemes were designed to investigate the effect of the thioether bonds at the C-14 side chain on the antimicrobial activity of the pleuromutilin derivative. The oxygen atom at the C-22 position of the pleuromutilin was replaced by the nitrogen atom to furnish compounds **12b**~**28b** in Figure 1. MIC values of most compounds in Figure 1 were not found (higher than 64 μg/mL), while compound **12b** exhibited a better antibacterial effect against MRSA (MIC = 8 μg/mL). In Figure 2, the oxygen atom at the C-22 position of the pleuromutilin was replaced by the sulfur atom, namely the sulphydryl at 1 position on 4-aminothiophenol, resulting in the formation of the thioether bond. As outlined in Table 2, the MIC of compound **32** against MRSA was 0.0015. Compared with Figure 1, compounds **12c**, **19c** and **22c** displayed enhanced antibacterial activity with the MIC values of 0.25 μg/mL against MRSA. The introduction of 4-aminothiophenol caused the formation of a sulfide structure at the C22 position, the antibacterial activity of the derivatives was significantly promoted. This is consistent with our previous findings that the thioether bond has a positive influence on the antimicrobial effect of pleuromutilin derivatives [31,32,33,34,35]. Furthermore, the substituents at the N1 position of the pyrazolo[3,4-*d*]pyrimidine ring contained aromatic rings with different substituents. A variety of electron-donating (methyl, ethyl) and electron-withdrawing (fluoro, chlorine, etc.) groups were introduced on the aromatic rings. The meta-methyl-substituted compound **12c** (0.25 μg/mL) showed superior antibacterial activities than compounds **13c**~**17c** against MRSA. Compounds **18c**, **20c** and **21c** with ortho-fluoro-substituted benzene derivatives exhibited much less potency compared to compounds **19c** and **22c** in this series. Furthermore, compound **23c**, bearing a trifluoromethoxy group in the 4-position, showed activity comparable to tiamulin (MIC was 0.5 μg/mL) against MRSA. Generally, the ortho-position appeared to have less positive influence on the antibacterial activity of these pleuromutilin derivatives bearing a phenyl group.

This may be due to the pyrazolo [3,4-*d*] pyrimidine ring and their substituent groups forming a rigid planar structure. There were a large number of hydrogen bond donors, such as N, O and F. This may prevent the extension of the C14 side chain of pleuromutilin to PTC of 50S ribosomes, because it does not have a flexible conformation. At the same time, a large number of hydrogen bond donors in their structure must be disturbed by environmental substances when the compounds produce effects, violating the Lipinski rules. Thus, the introduction of this pyrimidine ring at the C14 side chain of pleuromutilin may not be a good modification strategy.

MBC refers to the minimum drug concentration required to kill 99.9% of the test microorganisms [33]. The MBC/MIC ratios of compounds **12c**, **19c** and **22c** against MRSA were ≤2. According to previous studies, an antimicrobial drug can be considered bactericidal when the MBC/MIC is ≤4 [36]. Thus, the antimicrobial activity of compounds **12c**, **19c** and **22c** was investigated for in-depth study.

Time–kill kinetic assays were performed, which were used to investigate the in vitro bactericidal kinetic effect of compounds **12c**, **19c**, **22c** and tiamulin. The results are presented in Figure 2.

The test compounds displayed a significant inhibitory effect against MRSA at 1 × MIC. After 24 h incubation, compound **19c** and tiamulin showed bactericidal effects, killing 99.9% of MRSA (−3.57 log_10_ CFU/mL and −3.78 log_10_ CFU/mL reduction, respectively) at 2 × MIC. After incubation for 24 h, compounds **12c** and **22c** displayed bactericidal effects, killing 99.99% of MRSA (−4.51 log_10_ CFU/mL and −4.47 log_10_ CFU/mL reduction, respectively) at 2 × MIC. Compared with tiamulin, compounds **12c** and **22c** manifested faster bactericidal kinetics against MRSA. Compounds **12c**, **19c** and **22c** induced MRSA killing significantly at 4 × MIC (−5.76 log_10_ CFU/mL, −5.16 log_10_ CFU/mL and −5.79 log_10_ CFU/mL reduction, respectively). However, after the test compounds reached a certain concentration, the bactericidal effect did not significantly increase. The results indicated that compounds **12c**, **19c**, **22c** and tiamulin are time-dependent drugs rather than concentration-dependent drugs. In clinical practice, multiple or continuous intravenous time-dependent antimicrobial agents can achieve better therapeutic results [37].

PAE has been considered as pharmacodynamic assistance which provides a reference for the rational formulation of dosing regimen [38]. PAE refers to the temporary suppression of bacterial growth following transient antibiotic treatment [39]. PAE assays were conducted for compounds **12c** and **22c**. The results were shown in Table 3, and the bacterial growth kinetics curves were exhibited in Figure 3.

PAE is calculated from growth curves on the difference in time required for the number (1 log_10_ CFU/mL) of drug-exposed and unexposed microbes [40]. Following exposure to compounds **12c** and **22c** at the concentration of 2 × MIC for 1 h, the corresponding PAE values were 0.37 and 0.92 h, respectively. After exposure for 1 h, the PAE values of compounds **12c** and **22c** were 0.43 and 0.93 h at 4 × MIC. Correlatively, at the concentration of 2 × MIC, the PAE values of compounds **12c**, **22c** and tiamulin after exposure for 2 h were 1.72, 2.18 and 1.23 h, while at the concentration of 4 × MIC, corresponding PAE values were 1.73, 2.68 and 1.31 h. This suggests that compounds **12c** and **22c** induced longer PAEs against MRSA than tiamulin. The PAE values of compounds **12c** and **22c** increased with increasing incubation time. Compound **22c** showed the longest PAE.

Results obtained from the PAE assays showed that compounds **12c** and **22c** had been speculated to possess longer administration intervals than tiamulin. There is still a certain antibacterial effect as the drug concentration drops below the MIC. In clinical medication, pharmacodynamic parameters such as PAE, MIC and MBC can be used as references to evaluate the efficacy of antimicrobial agents. PAE provides a theoretical basis for the adjustment of the time interval to reduce adverse drug effects [41].

### 2.3. Cytotoxicity Assay

The presence of compounds can influence cellular basic physiological processes, suppress proliferation and even reduce cell survival, etc. In the community, MRSA generally causes respiratory infection [42]. Therefore, cytotoxicity was evaluated to explore the effect of compounds **12c** and **22c** on the viability of the respiratory tract cells 16-HBE (human bronchial epithelial cell line). RAW 264.7 (mouse peritoneal macrophage cell line) and Caco-2 (human epithelial colorectal adenocarcinoma cell line) were also used to evaluate the cytotoxicity of compounds **12c** and **22c** by MTT assay.

As shown in Figure 4, compound **22c** displayed slight inhibition on the viability of RAW 264.7 cells at the concentration of 8 µg/mL. Compounds **12c** and **22c** did not affect the viability of Caco-2 and 16-HBE cells at the concentrations of 1~8 μg/mL. The study indicated that compounds **12c** and **22c** possessed a safety profile towards RAW 264.7 cells, Caco-2 and 16-HBE cells at higher doses.

### 2.4. Neutropenic Murine Thigh Infection Model

Compounds **12c** and **22c** possessed superior in vitro activity against MRSA and had been identified to be non-cytotoxic on RAW 264.7, Caco-2 and 16-HBE cells at high doses. Therefore, the in vivo efficacy of compounds **12c** and **22c** were assessed using the murine neutropenic thigh infection model. Mice groups treated with saline and tiamulin were chosen as the negative and positive controls. The results are shown in Figure 5.

Compared to the growth control group, tiamulin could reduce MRSA load (−0.56 log_10_ CFU/mL) against MRSA in thigh muscle (*p* < 0.0001, n = 6/group). Compounds **12c** and **22c** at the same dose could reduce the MRSA load (−0.79 log_10_ CFU/mL and −0.93 log_10_ CFU/mL, respectively) in thighs compared with the blank control group (*p* < 0.0001, n = 6/group, both). Compounds **12c** and **22c** displayed more effective bactericidal activity than tiamulin in reducing MRSA load in thigh-infected mice (*p* = 0.0009 and 0.0001, respectively). This indicates preliminarily that compound **22c** had a better in vivo anti-MRSA effect than compound **12c** and tiamulin in the neutropenic murine thigh infection model, which could be used as a drug candidate against MRSA.

### 2.5. Molecular Docking Study

To predict the binding conformations of compound **22c** to 50S ribosomes, molecular docking experiments were conducted [43]. The crystal structure (PDB: 1XBP) was obtained from the RCSB Protein Data Bank [44]. The validation of the docking scheme was performed by evaluating the quantitatively root mean square deviation (RMSD) of atom positions between the docking pose (the test compound to 1XBP) and X-ray crystallographic conformation (tiamulin to 1XBP). The quality of protein–ligand interactions can be expressed to some extent by the ligand efficiency (LE), the average binding free energy per non-hydrogen (or heavy) atom of the ligand [44]. The RMSD of compound **22c** was 0.921 Å. Compounds **22c** and tiamulin showed similar binding modes with 50S ribosomes, which were presented in Figure 6a.

The binding free energy of compound **22c** with 50S ribosome was −7.11 kcal/mol. As shown in Figure 6b, four strong hydrogen bonds were found through the interaction of compound **22c** and the 50S ribosome, namely 3-O and G2044 (distance: 2.2 Å, 2.2 Å), 11-O and G2484 (distance: 2.7 Å) and 21-O and G2482 (distance: 2.4 Å). The result predicted the binding mode of compound **22c** to the 50S ribosome to indicate that they might have a good affinity.

## 3. Experimental Section

### 3.1. Materials

Pleuromutilin (>90% pure) was purchased from Great Enjoyhood Biochemical Co., Ltd., (Daying, China). Urea, sodium methanolatea and hydrochloric acid were purchased from Titan Technology Co., Ltd., (Shanghai, China). Diethyl malonate and hydrazine derivatives were purchased from Bide Technology Co., Ltd., (Shanghai, China). The other analytical-grade solvents were purchased from Guangzhou General Reagent Factory (Guangzhou, China). Chromatographic purification was carried out on silica gel columns (200–300 mesh, Branch of Qingdao Haiyang Chemical Co., Ltd., Shandong, China). ^1^H NMR and ^13^C NMR spectra were measured on a Bruker AV-600 spectrometer in chloroform-*d* or DMSO*-d_6_*. Tetramethylsilane was used as the internal standard. Chemical shift values (δ) were indicated as ppm. High-resolution mass spectrometry was performed using a Thermo Scientific Q Exactive Focus Orbitrap LC-MS/MS with an electrospray ionization (ESI) source.

### 3.2. Synthesis

Two series of novel pleuromutilin derivatives containing 6-chloro-4-amino-1-R-1*H*-pyrazolo[3,4-*d*]pyrimidine were synthesized. The general synthetic routes are illustrated in Figure 1 and Figure 2.

#### 3.2.1. 4,6-Dichloro-1-R-1*H*-pyrazolo[3,4-*d*]pyrimidine (**12a**~**28a**)

The raw material for synthesizing barbituric acid (compound **10**) was urea (compound **9**). Urea powder (6 g, 100 mmol) was dissolved in EtOH (50 mL), and then diethyl malonate (16 g, 120 mmol) was added to the solution. The reaction mixture was incubated with sodium methanolate (6.48 g, 120 mmol) for 48 h at 65 °C. The solution was acidified to pH 1~2, cooled down, crystallized and recrystallized to obtain pure barbituric acid. The Vilsmeier–Haack reaction of barbituric acid provided the product 2,4,6-trichloropyrimidine-5-carbaldehyde (compound **11**). Phosphorus oxychloride (17.9 g, 117 mmol) was added into a 3-neck boiling flask at −10 °C, then N,N-dimethylformamide (15 mL) was dropped slowly into Phosphorus oxychloride, and then stirred for 1 h. Afterwards, Barbituric acid (5 g, 39 mmol) was added to the mixture and stirred for 48 h at 100 °C. After the reaction finished, the solution was slowly added into the ice water, a large number of solid precipitated out and then filtered, and the yield of compound 11 was 92%. The pyrazole ring was closed by treatment with hydrazine derivatives. Hydrazine derivatives (5.64 mmol) and compound 11 (1 g, 4.7 mmol) were dissolved in EtOH (10 mL) and then stirred for 2 h at −20 °C under alkaline conditions. After the reaction was completed, a large amount of solid was precipitated from the reaction solution. The crude products were purified using EtOH to obtain compounds **12a**~**28a**. Yield: 83%~98%.

#### 3.2.2. 22-Amino-deoxypleuromutilin (**31**)

4-methylbenzene-1-sulfonyl chloride (5.6 g, 29.2 mmol) and pleuromutilin (10 g, 26.5 mmol) were dissolved in acetonitrile (50 mL), then sodium hydroxide granules (3 g, 52.84 mmol) were dissolved in water (15 mL) and dropped slowly into the above mixture solution and stirred in an ice bath for 3 h. The mixture was then vacuum-evaporated, extracted with 30 mL of dichloromethane and washed with water (30 mL). The organic layer was dried over anhydrous Na_2_SO**_4_** and filtered. Afterwards, isopropanol (50 mL) was added, and the mixture was heated at 70 °C until the solid was completely dissolved. Stewing the solution at room temperature for 1–2 h, the white solid precipitated and the white solid was collected (compound **29**), yield: 95%.

Compound **29** (1 g, 1.88 mmol) and sodium azide (0.37 g, 5.65 mmol) were added to 10 mL of acetone and 5 mL of water, respectively. The two solutions were mixed under continuous stirring and heated at 80 °C for 4 h. The mixture was then vacuum-evaporated, extracted with 30 mL of dichloromethane and washed with water (30 mL). The organic layer was dried over anhydrous Na_2_SO**_4_**, filtered and concentrated under reduced pressure to obtain the product compound **30**, yield: 93%. Compound **30** (10 g, 25.67 mmol) and triphenylphosphine (7.14 g, 28.24 mmol) were dissolved in THF (80 mL) and H_2_O (20 mL) solution. The mixture was maintained in an ice bath for 2 h, then washed with dichloromethane (30 mL) and water (50 mL) 3 times. The organic phase was dried over anhydrous Na_2_SO_4_ and evaporated in vacuum. The crude product was purified by column chromatography (dichloromethane: methanol = 200:1) using silica gel to obtain compound **31**.

#### 3.2.3. 22-(4-Amino-phenylsulfanyl)deoxypleuromutilin (**32**)

Compound **29** (10 g, 18.8 mmol) was dissolved in dichloromethane (100 mL), to which 4-aminothiophenol (2.5 g, 20 mmol) was added. Afterwards, 20% aqueous NaOH (10 mL) was added dropwise to the mixture and allowed to stir for 2 h at 70 °C. The crude product was purified by column chromatography (dichloromethane: methanol = 200:1) using silica gel to obtain compound **32**. Yield: 79%.

#### 3.2.4. General Procedure for the Synthesis of Compounds **12b**~**28b** and **12c**~**28c**

Compounds **12a**~**28a** (0.25 mmol) and compound **31** (1.28 g, 2.65 mmol) were added to tetrahydrofuran (15 mL). Triethylamine (0.001 mmol) was added as a catalyst. The reaction proceeded for 2~12 h at room temperature. After completion of the reaction, the crude product was obtained by filtration. Compounds **12b**~**28b** were purified by silica gel column chromatography (dichloromethane: methanol = 200:1). Compounds **12c**~**28c** were obtained by subjecting compounds **12a**~**28a** and compound **32** to the same procedures.

#### 3.2.5. 22-[(6-Chloro-1-(3-methylphenyl)-1*H*-pyrazolo[3,4-*d*]pyrimidine-4-yl)amino]-22-deoxypleuromutilin (**12b**)

Yellow powder; yield: 79%; melting point: 97–99 °C; ^1^H NMR (600 MHz, Chloroform-*d*) δ 9.46 (1 H, s), 8.10 (1 H, s), 7.17 (1 H, t, *J* = 7.8 Hz), 7.01 (1 H, s), 6.79 (2 H, m), 6.52 (1 H, dd, *J* = 17.4, 11.0 Hz, H19), 5.95 (1 H, d, *J* = 8.5 Hz, H14), 5.29–5.18 (2 H, m, H20), 4.44–4.18 (2 H, m, H22), 3.39 (1 H, d, *J* = 6.5 Hz, H11), 2.38 (s, 3H), 2.36–2.06 (5 H, m, H2, H4, H10, 11-OH), 1.82–1.49 (6 H, m, H1, H6, H7, H8), 1.47 (3 H, s, H15), 1.43 (2 H, m, H13), 1.20 (3 H, s, H18), 1.15 (1 H, m, H8), 0.91 (3 H, d, *J* = 7.0 Hz, H17), 0.80 (3 H, d, *J* = 7.0 Hz, H16). ^13^C NMR (151 MHz, Chloroform-*d*) δ 216.98 (C3), 168.80 (C21), 159.47, 156.63, 156.31, 142.94, 139.50, 138.82 (C19), 132.32, 129.23, 122.07, 117.58 (C20), 113.27, 109.73, 107.12, 74.62 (C11), 70.06 (C14), 58.16 (C4), 45.49 (C9), 44.75 (C6), 44.37 (C13), 44.02 (C12), 41.9 (C5), 36.71 (C10), 36.20 (C2), 34.46 (C22), 30.44 (C8), 26.90 (C7), 26.45 (C18), 24.89 (C1), 21.57, 16.87 (C16), 14.84 (C15), 11.50 (C17). HR-MS (ESI): Calcd for C_34_H_42_ClN_5_O**_4_** (M + Cl^−^): 654.2619; Found: 654.2618.

#### 3.2.6. 22-[(6-Chloro-1-(4-methylphenyl)-1*H*-pyrazolo[3,4-*d*]pyrimidine-4-yl)amino]-22-deoxypleuromutilin (**13b**)

Yellow powder; yield: 62%; melting point: 93–95 °C; ^1^H NMR (600 MHz, Chloroform-*d*) δ 9.47 (1 H, s), 8.06 (1 H, s), 7.07 (2 H, d, *J* = 8.1 Hz), 6.97–6.85 (2 H, m), 6.54 (1 H, dd, *J* = 17.4, 11.0 Hz, H19), 5.91 (1 H, d, *J* = 8.5 Hz, H14), 5.37–5.17 (2 H, m, H20), 4.41–4.20 (2 H, m, H22), 3.40 (1 H, d, *J* = 6.5 Hz, H11), 2.38–2.12 (8 H, m, H2, H4, H10, 11-OH, H34), 1.83–1.48 (6 H, m, H1, H6, H7, H8), 1.47 (3 H, s, H15), 1.43 (2 H, m, H13), 1.21 (3 H, m, H18), 1.16 (1 H, m, H8), 0.92 (3 H, d, *J* = 7.0 Hz, H17), 0.81 (3 H, d, *J* = 7.1 Hz, H16). ^13^C NMR (151 MHz, Chloroform-*d*) δ 217.01 (C3), 168.63 (C21), 159.23, 157.89, 157.43, 146.47 138.76, 135.98, 132.58, 117.40 (C20), 115.42, 106.26, 74.61 (C11), 70.28 (C14), 58.16 (C4), 45.49 (C9), 44.78 (C13), 44.62 (C6), 43.96 (C12), 41.87 (C5), 36.72 (C10), 36.04 (C2), 34.46 (C22), 30.49 (C8), 26.71 (C7), 26.15 (C18), 24.87 (C1), 18.44, 16.82 (C16), 14.82 (C15), 11.46 (C17). HR-MS (ESI): Calcd for C_34_H_42_ClN_5_O_4_ (M + Cl^−^): 654.2619; Found: 654.2617.

#### 3.2.7. 22-[(6-Chloro-1-(3,4-dimethylphenyl)-1*H*-pyrazolo[3,4-*d*]pyrimidine-4-yl)amino]-22-deoxypleuromutilin (**14b**)

Yellow powder; yield:72%; melting point: 99–103 °C; ^1^H NMR (600 MHz, Chloroform-*d*) δ 9.48 (1 H, s), 8.04 (1 H, s), 7.02 (1 H, d, *J* = 8.1 Hz), 6.96 (1 H, s), 6.75 (1 H, d, *J* = 9.9 Hz), 6.53 (1 H, dd, *J* = 17.4, 11.0 Hz, H19), 5.94 (1 H, d, *J* = 8.5 Hz, H14), 5.28–5.18 (2 H, m, H20), 4.42–4.24 (2 H, m, H22), 3.40 (1 H, d, *J* = 6.4 Hz, H11), 2.31–2.14 (11 H, m, H2, H4, H10, 11-OH, H34, H35), 1.75–1.55 (6 H, m, H1, H6, H7, H8), 1.47 (3 H, s, H15), 1.42 (2 H, m, H13), 1.20 (3 H, m, H18), 1.16 (1 H, m, H8), 0.92 (3 H, d, *J* = 7.0 Hz, H17), 0.80 (3 H, d, *J* = 7.0 Hz, H16). ^13^C NMR (151 MHz, DMSO-*d*_6_) δ 217.60 (C3), 168.41 (C21), 159.69, 155.53, 154.94, 142.09, 141.28, 137.63 (C19), 131.43, 130.59, 128.13, 115.81 (C20), 113.95, 110.47, 107.93, 73.07 (C11), 70.52 (C14), 57.64 (C4), 45.45 (C9), 44.57 (C6), 43.87 (C12), 41.98 (C5), 40.53 (C13), 37.00 (C10), 36.77 (C2), 34.46 (C22), 30.61 (C8), 29.02 (C7), 27.02 (C18), 24.95 (C1), 19.98, 19.12, 16.55 (C16), 14.80 (C15), 11.96 (C17). HR-MS (ESI): Calcd for C_35_H_44_ClN_5_O_4_ (M + Cl^−^): 668.2776; Found: 668.2778.

#### 3.2.8. 22-[(6-Chloro-1-(3,5-dimethylphenyl)-1*H*-pyrazolo[3,4-*d*]pyrimidine-4-yl)amino]-22-deoxypleuromutilin (**15b**)

Yellow powder; yield: 66%; melting point: 95–100 °C; ^1^H NMR (600 MHz, DMSO-*d*_6_) δ 9.48 (1 H, s), 8.3 (1 H, s) 6.81 (2 H, s), 6.50 (1 H, s), 6.17 (1 H, dd, *J* = 17.7, 11.2 Hz, H19), 5.72 (1 H, d, *J* = 8.2 Hz, H14), 5.03 (2 H, m, H20), 4.36 (2 H, m, H22), 3.44 (1 H, m, H11), 2.43 (1 H, s, 11-OH), 2.26 (6 H, s), 2.20–2.01 (4 H, m, H2, H4, H10), 1.69–1.20 (11 H, m, H1, H6, H7, H8, H13, H15), 1.06 (4 H, m, H8, H18), 0.83 (3 H, d, *J* = 6.9 Hz, H17), 0.65 (3 H, d, *J* = 6.3 Hz, H16). ^13^C NMR (151 MHz, Chloroform-*d*) δ 216.84 (C3), 168.49 (C21), 159.37, 156.73, 156.41, 142.94, 139.28, 138.83 (C19), 132.25, 123.18, 117.55 (C20), 110.73, 107.03, 74.60 (C11), 69.82 (C14), 58.17 (C4), 45.47 (C9), 44.75 (C13), 44.54 (C6), 43.92 (C12), 41.88 (C5), 36.70 (C10), 36.28 (C2), 34.44 (C22), 30.45 (C8), 26.82 (C7), 26.37 (C18), 24.90 (C1), 21.40, 16.79 (C16), 14.82 (C15), 11.40 (C17). HR-MS (ESI): Calcd for C_35_H_44_ClN_5_O_4_ (M + Cl^−^): 668.2776; Found: 668.2773.

#### 3.2.9. 22-[(6-Chloro-1-(4-ethylphenyl)-1*H*-pyrazolo[3,4-*d*]pyrimidine-4-yl)amino]-22-deoxypleuromutilin (**16b**)

Yellow powder; yield: 61%; melting point: 101–103 °C; ^1^H NMR (600 MHz, Chloroform-*d*) δ 9.49 (1 H, s), 8.07 (1 H, s), 7.11 (d, *J* = 8.4 Hz, 2H), 6.96 (d, *J* = 8.4 Hz, 2H), 6.55 (1 H, dd, *J* = 17.4, 11.0 Hz, H19), 5.92 (1 H, d, *J* = 8.5 Hz, H14), 5.31 (1 H, d, *J* = 11.0 Hz, H20), 5.24 (1 H, d, *J* = 18.7 Hz, H20), 4.43–4.22 (2 H, m, H22), 3.40 (1 H, d, *J* = 6.5 Hz, H11), 2.61 (2 H, q, *J* = 7.6 Hz), 2.39–2.12 (5 H, m, H2, H4, H10, 11-OH), 1.83–1.49 (6 H, m, H1, H6, H7, H8), 1.47 (3 H, s, H15), 1.44 (2 H, m, H13), 1.23 (t, *J* = 7.6 Hz, 3H), 1.21 (3 H, m, H18), 1.16 (1 H, m, H8), 0.91 (3 H, d, *J* = 7.0 Hz, H17), 0.81 (3 H, d, *J* = 7.0 Hz, H16). ^13^C NMR (151 MHz, Chloroform-*d*) δ 216.96 (C3), 168.71 (C21), 159.54, 156.51, 156.23, 140.92, 138.76 (C19), 137.09, 132.04, 128.71, 117.56 (C20), 112.75, 107.16, 74.61 (C11), 70.05 (C14), 58.13 (C4), 45.47 (C9), 44.68 (C13), 44.27 (C6), 44.04 (C12), 41.88 (C5), 36.70 (C10), 36.12 (C2), 34.45 (C22), 30.43 (C8), 28.09, 26.93 (C7), 26.40 (C18), 24.86 (C1), 16.87 (C16), 15.65, 14.82 (C15), 11.53 (C17). HR-MS (ESI): Calcd for C_35_H_44_ClN_5_O_4_ (M + Cl^−^): 668.2776; Found: 668.2773.

#### 3.2.10. 22-[(6-Chloro-1-(3-methoxyphenyl)-1*H*-pyrazolo[3,4-*d*]pyrimidine-4-yl)amino]-22-deoxypleuromutilin (**17b**)

Yellow powder; yield: 72%; melting point: 99–104 °C; ^1^H NMR (600 MHz, Chloroform*-d*) δ 9.41 (1 H, t, *J* = 5.1 Hz), 8.27 (1 H, s), 8.09 (1 H, s), 7.18 (1 H, t, *J* = 8.1 Hz), 6.61 (1 H, dd, *J* = 8.0, 1.6 Hz), 6.57 (1 H, t, *J* = 2.1 Hz), 6.53 (1 H, dd, *J* = 17.4, 11.0 Hz), 6.49 (1 H, dd, *J* = 8.1, 2.1 Hz, H19), 5.89 (1 H, d, *J* = 8.5 Hz, H14), 5.32–5.15 (2 H, m, H20), 4.44–4.22 (2 H, m, H22), 3.81 (3 H, s), 3.39 (1 H, d, *J* = 6.5 Hz, H11), 2.38–2.09 (5 H, m, H1, H6, H7, 11-OH), 1.81–1.496 (6 H, m, H1, H7, H6, H8), 1.47 (3 H, s, H15), 1.43 (2 H, m, H13), 1.20 (3 H, s, H18), 1.16 (1 H, m, H8), 0.91 (3 H, d, *J* = 7.0 Hz, H17), 0.80 (3 H, d, *J* = 7.1 Hz, H16). ^13^C NMR (151 MHz, Chloroform-*d*) δ 216.84 (C3), 168.36 (C21), 159.46, 156.66, 156.39, 141.05, 138.80 (C19), 137.88, 132.04, 130.39, 129.52, 117.53 (C20), 114.42, 110.41, 107.07, 74.60 (C11), 69.85 (C14), 58.15 (C4), 45.48 (C9), 44.72 (C13), 44.48 (C6), 43.97 (C12), 41.88 (C5), 36.70 (C10), 36.21 (C2), 34.44 (C22), 30.46 (C8), 26.85 (C7), 26.36 (C18), 24.88 (C1), 19.88, 19.03, 16.82 (C16), 14.81 (C15), 11.47 (C17). HR-MS (ESI): Calcd for C_34_H_42_ClN_5_O_5_ (M + Cl^−^): 670.2568; Found: 670.2569.

#### 3.2.11. 22-[(6-Chloro-1-(2-fluorophenyl)-1*H*-pyrazolo[3,4-*d*]pyrimidine-4-yl)amino]-22-deoxypleuromutilin (**18b**)

Yellow powder; yield: 52%; melting point: 107–110 °C; ^1^H NMR (600 MHz, Chloroform*-d*) δ 9.42 (1 H, t, *J* = 4.6 Hz), 8.28 (1 H, s), 7.68 (1 H, t, *J* = 8.8 Hz), 7.14 (1 H, t, *J* = 7.7 Hz), 7.07 (1 H, dd, *J* = 11.7, 8.2 Hz), 6.90 (1 H, q, *J* = 7.1, 6.6 Hz), 6.54 (1 H, dd, *J* = 17.4, 11.0 Hz, H19), 5.90 (1 H, d, *J* = 8.5 Hz, H14), 5.32–5.15 (2 H, m, H20), 4.44–4.23 (2 H, m, H22), 3.39 (1 H, d, *J* = 6.5 Hz, H11), 2.37–2.10 (5 H, m, H2, H4, H10, 11-OH), 1.80–1.54 (6 H, m, H1, H6, H7, H8), 1.47 (3 H, s, H15), 1.41 (2 H, m, H13), 1.20 (3 H, m, H18), 1.17 (1 H, m, H8), 0.92 (3 H, d, *J* = 7.0 Hz, H17), 0.78 (3 H, d, *J* = 7.1 Hz, H16). ^13^C NMR (151 MHz, Chloroform-*d*) δ 216.79 (C3), 168.36 (C21), 159.50, 157.58, 157.12, 138.69 (C19), 135.02, 131.60, 124.93, 120.87, 120.82, 117.59 (C20), 115.21, 114.39, 106.59, 74.60 (C11), 70.05 (C14), 58.09 (C4), 45.46 (C9), 44.70 (C13), 44.49 (C6), 44.04 (C12), 41.87 (C5), 36.66 (C10), 36.12 (C2), 34.43 (C22), 30.42 (C8), 26.90 (C7), 26.40 (C18), 24.86 (C1), 16.83 (C16), 14.80 (C15), 11.56 (C17). HR-MS (ESI): Calcd for C_33_H_39_ClFN_5_O_4_ (M + Cl^−^): 658.2369; Found:658.2367.

#### 3.2.12. 22-[(6-Chloro-1-(4-fluorophenyl)-1*H*-pyrazolo[3,4-*d*]pyrimidine-4-yl)amino]-22-deoxypleuromutilin (**19b**)

Yellow powder; yield: 57%; melting point: 112–115 °C; ^1^H NMR (600 MHz, Chloroform*-d*) δ 9.40 (1 H, d, *J* = 4.5 Hz), 8.18 (1 H, s), 7.14 (2 H, dd, *J* = 8.9, 4.4 Hz), 7.03 (2 H, t, *J* = 8.6 Hz), 6.53 (1 H, dd, *J* = 17.4, 11.0 Hz, H19), 5.89 (1 H, d, *J* = 8.6 Hz, H14), 5.33–5.23 (2 H, m, H20), 4.40–4.23 (2 H, m, H22), 3.39 (1 H, dd, *J* = 6.5 Hz, H11), 2.41–2.10 (5 H, m, H2, H4, H10, 11-OH), 1.75–1.53 (6 H, m, H1, H6, H7, H8, H13), 1.46 (3 H, s, H15), 1.41 (2 H, m, H13), 1.20 (3 H, m, H18), 1.18 (1 H, m, H8), 0.93 (3 H, d, *J* = 7.0 Hz, H17), 0.77 (3 H, d, *J* = 7.1 Hz, H16). ^13^C NMR (151 MHz, Chloroform-*d*) δ 216.90 (C3), 168.44 (C21), 159.45, 157.09, 157.08, 139.53, 138.69 (C19), 133.06, 117.60 (C20), 116.01, 114.17, 106.85, 74.61 (C11), 70.11 (C14), 58.10 (C4), 45.48 (C9), 44.67 (C13), 44.47 (C6), 44.05 (C12), 41.89 (C5), 36.66 (C10), 36.14 (C2), 34.45 (C22), 30.41 (C8), 26.91 (C7), 26.42 (C18), 24.86 (C1), 16.85 (C16), 14.81 (C15), 11.55 (C17). HR-MS (ESI): Calcd for C_33_H_39_ClFN_5_O_4_ (M + Cl^−^): 658.2369; Found: 658.2369.

#### 3.2.13. 22-[(6-Chloro-1-(2,4-difluorophenyl)-1*H*-pyrazolo[3,4-*d*]pyrimidine-4-yl)amino]-22-deoxypleuromutilin (**20b**)

Yellow powder; yield: 68%; melting point: 123–125 °C; ^1^H NMR (600 MHz, Chloroform*-d*) δ 9.38 (1 H, s), 8.33 (1 H, s), 7.70–7.53 (1 H, m), 6.89–6.71 (2 H, m), 6.52 (1 H, dd, *J* = 17.5, 11.0 Hz, H19), 5.90 (1 H, d, *J* = 8.6 Hz, H14), 5.28–5.20 (2 H, m, H20), 4.33 (2 H, ddd, *J* = 69.8, 18.9, 4.8 Hz, H22), 3.41 (1 H, dd, *J* = 10.2, 6.5 Hz, H11), 2.38–2.12 (5H, m, H2, H4, H10, 11-OH), 1.94–1.49 (6 H, m, H1, H6, H7, H8), 1.49 (3 H, s, H15), 1.43 (2 H, m, H13), 1.20 (3 H, m, H18), 1.16 (1 H, m, H8), 0.93 (3 H, d, *J* = 7.0 Hz, H17), 0.79 (3 H, d, *J* = 7.0 Hz, H16). ^13^C NMR (151 MHz, Chloroform-*d*) δ 216.81 (C3), 168.37 (C21), 159.41, 157.36, 156.99, 141.68, 138.65 (C19), 133.66, 129.43, 126.03, 117.64 (C20), 114.26, 106.67, 74.59 (C11), 70.17 (C14), 58.08 (C4), 45.47 (C9), 44.64 (C6), 44.53 (C13), 44.05 (C12), 41.89 (C5), 36.64 (C10), 36.14 (C2), 34.45 (C22), 30.40 (C8), 26.92 (C7), 26.39 (C18), 24.84 (C1), 16.87 (C16), 14.81 (C15), 11.54 (C17). HR-MS (ESI): Calcd for C_33_H_38_ClF_2_N_5_O_4_ (M + Cl^−^):676.2274; Found:676.2274.

#### 3.2.14. 22-[(6-Chloro-1-(2,5-difluorophenyl)-1*H*-pyrazolo[3,4-*d*]pyrimidine-4-yl)amino]-22-deoxypleuromutilin (**21b**)

Yellow powder; yield: 62%; melting point: 121–125 °C; ^1^H NMR (600 MHz, Chloroform*-d*) δ 9.35 (1 H, s), 8.29 (1 H, s), 7.64–7.52 (1 H, m), 7.01–6.92 (1 H, m), 6.59 (1 H, dd, *J* = 17.3, 11.0 Hz), 6.55–6.47 (1 H, m, H18), 6.02 (1 H, d, *J* = 8.5 Hz, H14), 5.29–5.12 (2 H, m, H20), 4.33 (2 H, m, H22), 3.42–3.34 (1 H, m, H11), 2.46–2.11 (5 H, m, H2, H4, H10, 11- OH), 1.85–1.48 (6 H, m, H1, H6, H7, H8), 1.48 (3 H, s, H15), 1.40 (2 H, d, *J* = 16.1 Hz, H13), 1.19 (3 H, m, H18), 1.16 (1 H, m, H8), 0.92 (3 H, d, *J* = 7.0 Hz, H17), 0.78 (3 H, d, *J* = 7.1 Hz, H16). ^13^C NMR (151 MHz, Chloroform-*d*) δ 216.97 (C3), 168.77 (C21), 159.52, 156.45, 156.17, 140.72, 138.78 (C19), 131.98, 130.53, 129.89, 117.53 (C20), 112.63, 107.16, 74.61 (C11), 70.10 (C14), 58.11 (C4), 45.47 (C9), 44.67 (C6), 44.25 (C13), 44.06 (C12), 41.88 (C5), 36.69 (C10), 36.11 (C2), 34.46 (C22), 30.42 (C8), 26.93 (C7), 26.41 (C18), 24.85 (C1), 16.87 (C16), 14.82 (C15), 11.55 (C17). HR-MS (ESI): Calcd for C_33_H_38_ClF_2_N_5_O_4_ (M + Cl^−^):676.2274; Found:676.2274

#### 3.2.15. 22-[(6-Chloro-1-(3,4-difluorophenyl)-1*H*-pyrazolo[3,4-*d*]pyrimidine-4-yl)amino]-22-deoxypleuromutilin (**22b**)

Yellow powder; yield: 61%; melting point: 130–132 °C; ^1^H NMR (600 MHz, Chloroform*-d*) δ 9.49 (1 H, t, *J* = 4.6 Hz), 8.22 (1 H, s), 7.28–7.21 (1 H, m), 7.06 (1 H, q, *J* = 8.9 Hz\), 6.73 (1 H, d, *J* = 8.9 Hz), 6.53 (1 H, dd, *J* = 17.4, 11.0 Hz, H19), 5.95 (1 H, d, *J* = 8.5 Hz, H14), 5.29–5.16 (2 H, m, H20), 4.32 (2 H, m, H22), 3.32 (1 H, m, H11), 2.42–2.08 (5 H, m, H2, H4, H10, 11-OH), 1.86–1.46 (6 H, m, H1, H6, H7, H8), 1.46 (3 H, s, H15), 1.41 (2 H, m, H13), 1.19 (3 H, s, H18), 1.16 (1 H, m, H8), 0.94 (3 H, d, *J* = 7.0 Hz, H17), 0.77 (3 H, d, *J* = 7.0 Hz, H16). ^13^C NMR (151 MHz, Chloroform-*d*) δ 216.74 (C3), 168.40 (C21), 159.40, 157.68, 157.25, 139.33, 138.67 (C19), 135.26, 132.09, 128.34, 117.58 (C20), 115.11, 111.57, 111.42, 106.50, 74.59 (C11), 70.17 (C14), 58.08 (C4), 45.46 (C9), 44.67 (C13), 44.58 (C6), 44.05 (C12), 41.88 (C5), 36.62 (C10), 36.16 (C2), 34.43 (C22), 30.40 (C8), 26.91 (C7), 26.43 (C18), 24.85 (C1), 16.84 (C16), 14.80 (C15), 11.54 (C17). HR-MS (ESI): Calcd for C_33_H_38_ClF_2_N_5_O_4_ (M + Cl^−^):676.2274; Found: 676.2274.

#### 3.2.16. 22-[(6-Chloro-1-(4-(trifluoromethoxy)Phenyl)-1*H*-pyrazolo[3,4-*d*]pyrimidine-4-yl)amino]-22-deoxypleuromutilin (**23b**)

Yellow powder; yield: 62%; melting point: 132–125 °C; ^1^H NMR (600 MHz, Chloroform*-d*) δ 9.39 (1 H, s), 8.19 (1 H, s), 7.20-7.11 (4 H, m), 6.53 (1 H, dd, *J* = 17.5, 11.0 Hz, H19), 5.92 (1 H, d, *J* = 8.5 Hz, H14), 5.29-5.20 (2 H, m, H20), 4.43–4.24 (2 H, m, H22), 3.40 (1 H, d, *J* = 6.5 Hz, H11), 2.37–2.11 (5 H, m, H2, H4, H10, 11-OH), 1.84–1.51 (7 H, m, H1, H6, H7, H8), 1.48 (3 H, s, H15), 1.43 (2 H, m, H13), 1.21 (3 H, s, H18), 1.16 (1 H, m, H8), 0.93 (3 H, d, *J* = 7.2 Hz, H17), 0.78 (3 H, d, *J* = 7.0 Hz, H16). ^13^C NMR (151 MHz, Chloroform-*d*) δ 216.97 (C3), 168.62 (C21), 159.41, 157.29, 156.93, 143.14, 141.82, 138.64 (C19), 133.75, 122.42, 117.55 (C20), 113.61, 106.74, 74.60 (C11), 70.23 (C14), 58.13 (C4), 45.49 (C13), 44.66 (C6), 44.50 (C12), 44.04, 41.89 (C5), 36.67 (C10), 36.17 (C2), 34.46 (C22), 30.40 (C8), 26.90 (C7), 26.41 (C18), 24.86 (C1), 16.84 (C16), 14.83 (C15), 11.44 (C17). HR-MS (ESI): Calcd for C_34_H_39_ClF_3_N_5_O_5_ (M + Cl^−^): 724.2286; Found: 724.2290.

#### 3.2.17. 22-[(6-Chloro-1-(3-chlorophenyl)-1*H*-pyrazolo[3,4-*d*]pyrimidine-4-yl)amino]-22-deoxypleuromutilin (**24b**)

Yellow powder; yield: 69%; melting point: 117–119 °C; ^1^H NMR (600 MHz, Chloroform*-d*) δ 9.37 (1 H, s), 8.24 (1 H, s), 7.39 (1 H, s), 7.19 (1 H, t, *J* = 8.0 Hz), 6.91 (1 H, d, *J* = 8.9 Hz), 6.84 (1 H, d, *J* = 9.5 Hz\), 6.61 (1 H, dd, *J* = 17.4, 11.0 Hz, H19), 6.00 (1 H, d, *J* = 8.5 Hz, H14), 5.27–5.14 (2 H, m, H20), 4.32 (2 H, m, H22), 3.38 (1 H, d, *J* = 6.0 Hz, H11), 2.37–2.12 (5 H, m, H2, H4, H10, 11-OH), 1.67 (6 H, m, H1, H6, H7, H8), 1.48 (3 H, s, H15), 1.41 (2 H, m, H13), 1.19 (3 H, s, H18), 1.16 (1 H, m, H8), 0.91 (3 H, d, *J* = 7.0 Hz, H17), 0.80 (3 H, d, *J* = 7.0 Hz, H16). ^13^C NMR (151 MHz, Chloroform*-d*) δ 217.01 (C3), 168.50 (C21), 159.31, 157.39, 157.04, 144.17, 138.98 (C19), 135.66, 133.98, 130.28, 121.14, 117.45 (C20), 112.95, 111.10, 106.58, 74.60 (C11), 70.15 (C14), 58.16 (C4), 45.49 (C9), 44.76 (C13), 44.55 (C6), 43.98 (C12), 41.87 (C5), 36.74 (C10), 36.15 (C2), 34.47 (C22), 30.47 (C8), 26.82 (C7), 26.19 (C18), 24.87 (C1), 16.87 (C16), 14.82 (C15), 11.48 (C17). HR-MS (ESI): Calcd for C_33_H_39_Cl_2_N_5_O_4_ (M + Cl^−^): 676.2042; Found: 676.2042.

#### 3.2.18. 22-[(6-Chloro-1-(4-chlorophenyl)-1*H*-pyrazolo[3,4-*d*]pyrimidine-4-yl)amino]-22-deoxypleuromutilin (**25b**)

Yellow powder; yield: 66%; melting point: 107–111 °C, ^1^H NMR (600 MHz, Chloroform*-d*) δ 9.37 (1 H, s), 8.18 (1 H, s), 7.28 (2 H, d, *J* = 8.5Hz), 7.13 (2 H, d, *J* = 8.8 Hz), 6.54 (1 H, dd, *J* = 17.4, 11.0 Hz, H19), 5.90 (2 H, d, *J* = 8.6 Hz, H14), 5.32 (1 H, d, *J* = 11.3 Hz, H20), 5.24 (1 H, d, *J* = 17.4 Hz, H20), 4.47–4.18 (2 H, m, H22), 3.44–3.34 (1 H, m, H11), 2.40–2.10 (5 H, m, H2, H4, H10, 11-OH), 1.75–1.56 (7 H, m, H1, H6, H7, H8), 1.47 (3 H, s, H15), 1.42 (2 H, m, H13), 1.20 (3 H, d, *J* = 7.0 Hz, H18), 1.16 (1 H, m, H8), 0.93 (3 H, d, *J* = 7.0 Hz, H17), 0.77 (3 H, d, *J* = 7.1 Hz, H16). ^13^C NMR (151 MHz, Chloroform-*d*) δ 216.92 (C3), 168.45 (C21), 159.24, 157.53, 157.19, 144.44, 138.71 (C19), 134.07, 117.46 (C20), 108.20, 106.47, 102.79, 102.64, 74.62 (C11), 70.20 (C14), 58.14 (C4), 45.49 (C9), 44.63 (C13), 43.98 (C12), 41.88 (C5), 36.69 (C10), 36.10 (C2), 34.46 (C22), 30.46 (C8), 29.71 (C6), 26.77 (C7), 26.24 (C18), 24.87 (C1), 16.82 (C16), 14.81 (C15), 11.48 (C17). HR-MS (ESI): Calcd for C_33_H_39_Cl_2_N_5_O_4_ (M + Cl^−^): 676.2042; Found: 676.2042.

#### 3.2.19. 22-[(6-Chloro-1-(3,5-dichlorophenyl)-1*H*-pyrazolo[3,4-*d*]pyrimidine-4-yl)amino]-22-deoxypleuromutilin (**26b**)

Yellow powder; yield: 56%; melting point: 151–156 °C; ^1^H NMR (600 MHz, Chloroform*-d*) δ 9.33 (1 H, s), 8.21 (1 H, s), 7.22 (1 H, s), 6.94 (1 H, s), 6.64 (1 H, dd, *J* = 17.3, 11.0 Hz, H19), 6.06 (1 H, d, *J* = 8.4 Hz, H14), 5.28–5.10 (2 H, m, H20), 4.30 (2 H, m, H22), 3.41–3.33 (1 H, m, H11), 2.40–2.08 (5 H, m, H2, H4, H10, 11-OH), 1.89–1.58 (6 H, m, H1, H6, H7, H8), 1.47 (3 H, s, H15), 1.36 (2 H, m, H13), 1.18 (3 H, s, H18), 1.16 (1 H, m, H8), 0.90 (3 H, d, *J* = 7.0 Hz, H17), 0.77 (3 H, d, *J* = 7.0 Hz, H16). ^13^C NMR (151 MHz, Chloroform-*d*) δ 216.92 (C3), 168.40 (C21), 159.13, 157.56, 144.74, 139.11 (C19), 135.93, 135.58, 135.24, 121.03, 117.39 (C20), 111.50, 106.16, 74.58 (C11), 70.16 (C14), 58.18 (C4), 45.50 (C9), 44.80 (C13), 44.72 (C6), 43.91 (C12), 41.85 (C10), 36.74 (C10), 36.23 (C2), 34.46 (C22), 30.50 (C8), 26.75 (C7), 26.07 (C18), 24.89 (C1), 16.85 (C16), 14.82 (C15), 11.41 (C17). HR-MS (ESI): Calcd for C_33_H_38_Cl_3_N_5_O_4_ (M + Cl^−^): 710.1654; Found: 710.1656.

#### 3.2.20. 22-[(6-Chloro-1-(3-nitrophenyl)-1*H*-pyrazolo[3,4-*d*]pyrimidine-4-yl)amino]-22-deoxypleuromutilin (**27b**)

Yellow powder; yield: 69%; melting point: 131–135 °C; ^1^H NMR (600 MHz, Chloroform*-d*) δ 9.39 (1 H, s), 8.35 (1 H, s), 8.01 (1 H, s), 7.78 (2 H, d, *J* = 7.8 Hz), 7.44 (2 H, dt, *J* = 16.1, 8.3 Hz), 6.50 (1 H, dd, *J* = 17.4, 11.0 Hz, H19), 5.91 (1 H, d, *J* = 8.6 Hz, H14), 5.19–5.08 (2 H, m, H20), 4.41–4.26 (2 H, m, H22), 3.44–3.34 (1 H, m, H11), 2.41–2.09 (5 H, m, H2, H4, H10, 11-OH), 1.88–1.60 (6 H, m, H1, H6, H7, H8), 1.48 (3 H, s, H15), 1.40 (2 H, m, H13), 1.18 (3 H, m, H18), 1.14 (1 H, m, H8), 0.94 (3 H, d, *J* = 7.0 Hz, H17), 0.80 (3 H, d, *J* = 7.0 Hz, H16). ^13^C NMR (151 MHz, Chloroform-*d*) δ 216.91 (C3), 168.43 (C21), 159.42, 157.60, 149.65, 144.07, 138.89 (C19), 135.47, 130.15, 118.55, 117.13 (C20), 115.71, 107.61, 106.23, 98.57, 74.61 (C11), 70.55 (C14), 58.08 (C4), 45.50 (C9), 44.66 (C13), 44.51 (C6), 44.11 (C12), 41.90 (C5), 36.69 (C10), 35.98 (C2), 34.48 (C22), 30.45 (C8), 26.78 (C7), 26.24 (C18), 24.84 (C1), 16.95 (C16), 14.81 (C15), 11.48 (C17). HR-MS (ESI): Calcd for C_33_H_39_ClN_6_O_6_ (M + Cl^−^): 685.2314; Found: 685.2310.

#### 3.2.21. 22-[(6-Chloro-1-(naphthalen-2-yl)-1*H*-pyrazolo[3,4-*d*]pyrimidine-4-yl)amino]-22-deoxypleuromutilin (**28b**)

Yellow powder; yield:63%; melting point: 123–125 °C; ^1^H NMR (600 MHz, Chloroform*-d*) δ 9.55 (1 H, s), 8.21 (1 H, s), 7.92 (1 H, d, *J* = 8.2 Hz), 7.84 (1 H, s), 7.77 (2 H, t, *J* = 8.7 Hz), 7.36 (2 H, m, 7.3 Hz), 7.20 (1 H, dd, *J* = 8.8, 2.1 Hz), 6.54 (1 H, dd, *J* = 17.4, 11.0 Hz, H19), 6.01 (1 H, d, *J* = 8.5 Hz, H14), 5.18-4.98 (2 H, m, H20), 4.48–4.27 (2 H, m, H22), 3.40 (1 H, d, *J* = 6.4 Hz, H11), 2.48-2.14 (5 H, m, H2, H4, H10, 11-OH), 1.68–1.49 (6 H, m, H1, H6, H7, H8), 1.49 (3 H, s, H15), 1.42 (2 H, m, H13), 1.19 (1 H, m, H8), 1.17 (3 H, s, H16), 0.98 (3 H, d, *J* = 6.7 Hz, H17), 0.81 (3 H, d, *J* =6.8 Hz, H16). ^13^C NMR (151 MHz, Chloroform-*d*) δ 216.83 (C3), 168.43 (C21), 159.34, 157.13, 156.76, 140.42, 138.60 (C19), 134.81, 133.33, 129.38, 129.29, 127.77, 126.93, 126.54, 123.64, 117.63 (C20), 115.15, 107.83, 106.88, 74.61 (C11), 70.03 (C14), 58.14 (C4), 45.48(C9), 44.82 (C6), 44.74 (C13), 43.96 (C12), 41.91 (C5), 36.70 (C10), 36.26 (C2), 34.44 (C22), 30.51 (C8), 26.92 (C7), 26.38 (C18), 24.91 (C1), 16.84 (C16), 14.83 (C15), 11.75 (C17). HR-MS (ESI): Calcd for C_37_H_42_ClN_5_O_4_ (M + Cl^−^): 690.2619; Found: 690.2616.

#### 3.2.22. 22-[4-(6-Chloro-1-(3-methylphenyl)-1*H*-pyrazolo[3,4-*d*]pyrimidine-4-yl)amino-Phenylsulfanyl]-22-deoxypleuromutilin (**12c**)

Yellow powder; yield: 52%; melting point: 95–97 °C; ^1^H NMR (600 MHz, DMSO*-d*_6_) δ 11.42 (1 H, s), 8.35 (1 H, s), 7.66–7.60 (2 H, m), 7.47 (1 H, d, *J* = 8.2 Hz), 7.19 (1 H, t, *J* = 7.7 Hz), 6.86 (1 H, s), 6.75 (1 H, dd, *J* = 34.4, 7.4 Hz), 6.08–5.99 (1 H, m, H19), 5.51 (1 H, d, *J* = 8.3 Hz, H14), 4.96 (2 H, m, H20), 3.90–3.75 (2 H, m, H22), 3.37 (1 H, t, *J* = 5.7 Hz, H11), 2.29 (3 H, s), 2.22-1.99 (5 H, m, H2, H4, H10, 11-OH), 1.70–1.34 (5 H, m, H1, H6, H7), 1.33 (3 H, s, H15), 1.25 (3 H, m, H8, H13), 0.98 (4 H, m, H8, H18), 0.79 (3 H, d, *J* = 7.0 Hz, H17), 0.58 (3 H, d, *J* = 7.1 Hz, H16). ^13^C NMR (151 MHz, DMSO*-d_6_*) δ 217.59 (C3), 168.11 (C21), 157.33, 156.67, 154.91, 144.06, 141.17 (C19), 139.37, 136.40, 132.14, 131.26, 130.24, 129.87, 122.45, 121.73, 115.63 (C20), 113.12, 110.26, 108.50, 73.04 (C11), 70.23 (C14), 57.70 (C4), 45.40 (C9), 44.42 (C13), 44.14 (C12), 41.92 (C5), 36.83 (C6), 36.81 (C10), 36.20 (C2), 34.44 (C22), 30.55 (C8), 28.96 (C7), 27.04 (C18), 24.91 (C1), 21.79, 16.55 (C16), 14.98 (C15), 11.97 (C17). HR-MS (ESI): Calcd for C_40_H_46_ClN_5_O_4_S (M + Cl^−^): 762.2653; Found: 762.2654.

#### 3.2.23. 22-[4-(6-Chloro-1-(4-methylphenyl)-1*H*-pyrazolo[3,4-*d*]pyrimidine-4-yl)amino-phenylsulfanyl]-22-deoxypleuromutilin (**13c**)

Yellow powder; yield: 61%; melting point: 101–105 °C; ^1^H NMR (600 MHz, Chloroform*-d*) δ 11.26 (1 H, s), 8.29 (1 H, s), 7.64 (2 H, d, *J* = 8.6 Hz), 7.41 (2 H, d, *J* = 8.6 Hz), 7.12 (2 H, d, *J* = 8.1 Hz,), 6.90 (2 H, d, *J* = 8.3 Hz), 6.45–6.36 (1 H, m, H19), 5.74 (1 H, d, *J* = 8.5 Hz, H14), 5.31 (1 H, m, H20), 5.16 (1 H, d, *J* = 18.5 Hz, 1*H*20), 3.62-3.47 (2 H, m, H22), 3.32 (1 H, m, H11), 2.32 (s, 3H), 2.30–1.98 (5 H, m, H2, H4, H10, 11-OH), 1.78–1.43 (6 H, m, H1, H6, H7, H8, H13), 1.43 (3 H, s, H15), 1.35 (2 H, m, H13), 1.12 (4 H, m, H8, H18), 0.86 (3 H, d, *J* = 7.0 Hz, H17), 0.70 (3 H, d, *J* = 7.1 Hz, H16). ^13^C NMR (151 MHz, Chloroform-*d*) δ 217.16 (C3), 168.36 (C21), 157.10, 156.29, 140.71, 138.87 (C19), 137.13, 132.32, 131.63, 131.27, 130.29, 130.17, 121.99, 117.33 (C20), 113.08, 113.06, 107.41, 74.60 (C11), 69.64 (C14), 58.18 (C4), 45.43 (C9), 44.76 (C13), 43.87 (C12), 41.77 (C5), 37.77 (C6), 36.77 (C10), 35.97 (C2), 34.48 (C22), 30.41 (C8), 26.84 (C7), 26.37 (C18), 24.83 (C1), 20.66, 16.79 (C16), 14.89 (C15), 11.51 (C17). HR-MS (ESI): Calcd for C_40_H_46_ClN_5_O_4_S (M + Cl^−^): 762.2653; Found: 762.2654.

#### 3.2.24. 22-[4-(6-Chloro-1-(3,4-dimethylphenyl)-1*H*-pyrazolo[3,4-d]pyrimidine-4-yl)amino-Phenylsulfanyl]-22-deoxypleuromutilin (**14c**)

Yellow powder; yield: 58.58%; melting point: 103–105 °C; ^1^H NMR (600 MHz, Chloroform*-d*) δ 11.09 (1 H, s), 8.20 (1 H, s), 7.88 (1 H, d, *J* = 8.8 Hz), 7.54 (1 H, d, *J* = 9.1 Hz), 7.37 (1 H, t, *J* = 8.4 Hz), 7.16 (1 H, t, *J* = 8.2 Hz), 6.77 (2 H, s), 6.59 (1 H, s), 6.36 (1 H, dd, *J* = 17.4, 11.0 Hz, H19), 5.62 (1 H, d, *J* = 8.5 Hz, H14), 5.27 (1 H, d, *J* = 11.0 Hz, H20), 5.13 (1 H, d, *J* = 17.4 Hz, H20), 3.40–3.32 (2 H, m, H22), 3.29 (1 H, d, *J* = 6.5 Hz, H11), 2.25 (6 H, s), 2.18–1.89 (5 H, m, H2, H4, H10, 11-OH), 1.70–1.43 (6 H, m, H1, H6, H7, H8), 1.28 (3 H, s, H15), 1.26 (2 H, m, H13), 1.09 (4 H, m, H8, H18), 0.85 (3 H, d, *J* = 7.0 Hz, H17), 0.53 (3 H, d, *J* = 7.0 Hz, H16). ^13^C NMR (151 MHz, Chloroform-*d*) δ 217.05 (C3), 168.27 (C21), 157.51, 157.26, 140.96, 138.88, 138.12 (C19), 137.21, 132.25, 131.69, 130.72, 130.23, 122.05, 117.35 (C20), 114.57, 110.59, 107.37, 74.62 (C11), 69.60 (C14), 58.17 (C4), 45.44 (C9), 44.78 (C13), 43.881 (C12), 41.78 (C5), 37.86 (C6), 36.77 (C10), 35.99 (C2), 34.46 (C22), 30.43 (C8), 26.84 (C7), 26.31 (C18), 24.84 (C1), 20.14, 19.02, 16.78 (C16), 14.88 (C15), 11.50 (C17). HR-MS (ESI): Calcd for C_41_H_48_ClN_5_O_4_S (M + Cl^−^): 776.2810; Found: 776.2810.

#### 3.2.25. 22-[4-(6-Chloro-1-(3,5-dimethylphenyl)-1*H*-pyrazolo[3,4-*d*]pyrimidine-4-yl)amino-Phenylsulfanyl]-22-deoxypleuromutilin (**15c**)

Yellow powder; yield: 59%; melting point: 92–95 °C; ^1^H NMR (600 MHz, Chloroform*-d*) δ 11.36 (1 H, s), 8.22 (1 H, s), 7.73 (2 H, d, *J* = 8.6 Hz), 7.44 (2 H, d, *J* = 8.6 Hz), 6.68 (3 H, s), 6.50–6.37 (1 H, m, H19), 5.74 (1 H, d, *J* = 8.1 Hz, H14), 5.35–5.11 (2 H, m, H20), 3.62–3.51 (2 H, s, H22), 3.32 (1 H, m, H11), 2.33 (6 H, s), 2.21–2.02 (5 H, m, H2, H4, H10, 11-OH), 1.69–1.44 (6 H, m, H1, H6, H7, H8), 1.42 (3 H, s, H15), 1.35 (2 H, m, H13), 1.13 (1 H, m, H8), 1.11 (3 H, m, H18), 0.86 (3 H, d, *J* = 6.9 Hz, H17), 0.69 (3 H, d, *J* = 6.8 Hz, H16). ^13^C NMR (151 MHz, Chloroform-*d*) δ 217.04 (C3), 168.24 (C21), 157.65, 157.17, 156.57, 142.93, 139.63, 138.86 (C19), 137.23, 132.51, 131.68, 130.41, 123.76, 121.96, 117.36 (C20), 110.99, 107.32, 74.61 (C11), 69.60 (C14), 58.17 (C4), 45.44 (C9), 44.79 (C13), 43.88 (C12), 41.77 (C5), 37.89 (C6), 36.77 (C10), 35.99 (C2), 34.46 (C22), 30.43 (C8), 26.84 (C7), 26.30 (C18), 24.84 (C1), 21.53, 16.77 (C16), 14.88 (C15), 11.50 (C17). HR-MS (ESI): Calcd for C_41_H_48_ClN_5_O_4_S (M + Cl^−^): 776.2810; Found: 776.2812.

#### 3.2.26. 22-[4-(6-Chloro-1-(4-ethylphenyl)-1*H*-pyrazolo[3,4-*d*]pyrimidine-4-yl)amino-Phenylsulfanyl]-22-deoxypleuromutilin (**16c**)

Yellow powder; yield: 74%; melting point: 97–99 °C; ^1^H NMR (600 MHz, Chloroform*-d*) δ 11.26 (1 H, s), 8.23 (1 H, s), 7.67 (2 H, d, *J* = 8.6 Hz), 7.45 (2 H, d, *J* = 10.9 Hz), 7.19 (2 H, d, *J* = 7.7 Hz), 6.97 (2 H, d, *J* = 7.5 Hz), 6.48–6.37 (1 H, m, H19), 5.74 (1 H, d, *J* = 8.1 Hz, H14), 5.33 (1 H, d, *J* = 11.0 Hz, H20), 5.17 (1 H, d, *J* = 17.4 Hz, H20), 3.58 (2 H, d, *J* = 15.7 Hz, H22), 3.32 (1 H, m, H11), 2.65 (2 H, d, *J* = 7.4 Hz), 2.33–2.03 (5 H, m, H2, H4, H10, 11-OH), 1.78–1.43 (6 H, m, H1, H6, H7, H8), 1.43 (3 H, s, H15), 1.34 (2 H, m, H13), 1.24 (3 H, m), 1.12 (3 H, m, H18), 1.06 (1 H, m, H8), 0.86 (3 H, d, *J* = 7.0 Hz, H17), 0.70 (3 H, d, *J* = 6.9 Hz, H16). ^13^C NMR (151 MHz, Chloroform-*d*) δ 217.05 (C3), 168.31 (C21), 157.61, 157.26, 156.53, 140.85, 138.90, 138.01 (C19), 137.10, 132.55, 131.67, 130.48, 129.09, 122.19, 117.35 (C20), 113.16, 107.31, 74.62 (C11), 69.61 (C14), 58.18 (C4), 45.45 (C9), 44.79 (C13), 43.89 (C12), 41.79 (C5), 37.81 (C6), 36.78 (C10), 36.00 (C2), 34.46 (C22), 30.43 (C8), 28.14, 26.85 (C7), 26.33 (C18), 24.84 (C1), 16.78 (C16), 15.78, 14.89 (C15), 11.51 (C17). HR-MS (ESI): Calcd for C_41_H_48_ClN_5_O_4_S (M + Cl^−^): 776.2810; Found: 776.2814.

#### 3.2.27. 22-[4-(6-Chloro-1-(3-methoxyphenyl)-1*H*-pyrazolo[3,4-*d*]pyrimidine-4-yl)amino-Phenylsulfanyl]-22-deoxypleuromutilin (**17c**)

Yellow powder; yield: 73%; melting point: 97–102 °C; ^1^H NMR (600 MHz, Chloroform*-d*) δ 11.20 (1 H, s), 8.24 (1 H, s), 7.68 (2 H, d, *J* = 8.7 Hz), 7.45 (2 H, d, *J* = 8.7 Hz), 7.25 (1 H, t), 6.64–6.53 (3 H, m), 6.42 (1 H, dd, *J* = 17.4, 11.0 Hz, H19), 5.74 (1 H, d, *J* = 8.5 Hz, H14), 5.33 (1 H, dd, *J* = 11.0, 1.5 Hz, H20), 5.17 (1 H, dd, *J* = 17.4, 1*H*), 3.59–3.51 (2 H, m, H22), 3.35–3.29 (1 H, m, H11), 2.32–1.99 (5 H, m, H1, H6, H7, 11-OH), 1.77–1.46 (9 H, m, H1, H7, H6, H8, H40), 1.42 (3 H, s, H15), 1.39–1.27 (2H, m, H13), 1.11 (4 H, s, H8, H18), 0.86 (3 H, d, *J* = 7.0 Hz, H17), 0.70 (3 H, d, *J* = 7.1 Hz, H16). ^13^C NMR (151 MHz, Chloroform-*d*) δ 217.06 (C3), 168.26 (C21), 161.09, 157.93, 157.28, 156.79, 144.21, 138.87 (C19), 137.00, 133.24, 131.65, 130.66, 130.59, 122.23, 117.36 (C20), 107.13, 106.89, 105.67, 99.40, 74.62 (C11), 69.60 (C14), 58.17 (C4), 55.38, 45.44 (C9), 44.78 (C13), 43.88 (C12), 41.78 (C5), 37.81 (C6), 36.77 (C10), 35.99 (C2), 34.46 (C22), 30.43 (C8), 26.84 (C7), 26.31 (C18), 24.84 (C1), 16.77 (C16), 14.88 (C15), 11.50 (C17). HR-MS (ESI): Calcd for C_40_H_46_ClN_5_O_5_S (M + Cl^−^): 778.2602; Found: 778.2604.

#### 3.2.28. 22-[4-(6-Chloro-1-(2-fluorophenyl)-1*H*-pyrazolo[3,4-*d*]pyrimidine-4-yl)amino-Phenylsulfanyl]-22-deoxypleuromutilin (**18c**)

Yellow powder; yield: 68%; melting point: 123–127 °C; ^1^H NMR (600 MHz, Chloroform*-d*) δ 11.12 (1 H, s), 8.32 (1 H, s), 7.63 (2 H, d, *J* = 8.6 Hz), 7.42 (2 H, d, *J* = 8.6 Hz), 7.26 (1 H, d, *J* = 12.9 Hz), 7.21–7.06 (2 H, m), 6.93 (1 H, t, *J* = 10.9 Hz), 6.42 (1 H, dd, *J* = 17.4, 11.0 Hz, H19), 5.74 (1 H, d, *J* = 8.5 Hz, H14), 5.31 (1 H, m, H20), 5.16 (1 H, d, *J* = 18.6 Hz, H20), 3.62–3.45 (2 H, m, H22), 3.33 (1 H, t, *J* = 7.6 Hz, H11), 2.32–1.99 (5 H, m, H2, H4, H10, 11-OH), 1.83–1.44 (6 H, m, H1, H6, H7, H8), 1.43 (3 H, s, H15), 1.39–1.18 (2 H, m, H13), 1.12 (3 H, m, H18), 1.08 (1 H, m, H8), 0.86 (3 H, d, *J* = 7.0 Hz, H17), 0.70 (3 H, d, *J* = 7.1 Hz, H16). ^13^C NMR (151 MHz, Chloroform-*d*) δ 217.15 (C3), 168.30 (C21), 158.28, 157.28, 157.03, 150.84, 149.24, 138.90 (C19), 136.88, 135.25, 131.58, 130.68, 125.06, 122.14, 121.44, 117.27 (C20), 115.76, 115.64, 113.76, 106.92, 74.59 (C11), 69.65 (C14), 58.16 (C4), 45.43 (C9), 44.78 (C13), 43.88 (C12), 41.77 (C5), 37.68 (C6), 36.76 (C10), 35.97 (C2), 34.45 (C22), 30.40 (C8), 26.83 (C7), 26.35 (C18), 24.82 (C1), 16.78 (C16), 14.88 (C15), 11.49 (C17). HR-MS (ESI): Calcd for C_39_H_43_ClFN_5_O_4_S (M + Cl^−^): 766.2402; Found: 766.2405.

#### 3.2.29. 22-[4-(6-Chloro-1-(4-fluorophenyl)-1*H*-pyrazolo[3,4-*d*]pyrimidine-4-yl)amino-Phenylsulfanyl]-22-deoxypleuromutilin (**19c**)

Yellow powder; yield: 74%; melting point: 141–144 °C; ^1^H NMR (600 MHz, Chloroform*-d*) δ 11.19 (1 H, s), 8.64 (1 H, s), 7.60 (2 H, d, *J* = 8.7 Hz), 7.41 (d, *J* = 8.6 Hz, 2H), 7.04 (2 H, t, *J* = 8.5 Hz), 6.99-6.92 (2 H, m), 6.41 (1 H, dd, *J* = 17.4, 11.2 Hz, H19), 5.74 (1 H, d, *J* = 8.5 Hz, H14), 5.29 (1 H, m, H20), 5.16 (1 H, d, *J* = 18.6 Hz, H20), 3.57 (2 H, m, H22), 3.34 (1 H, t, *J* = 5.9 Hz, H11), 2.33–2.08 (5 H, m, H2, H4, H10, 11-OH), 1.77–1.44 (6 H, m, H1, H6, H7, H8), 1.43 (3 H, s, H15), 1.36 (2 H, m, H13), 1.12 (4 H, m, H8, H18), 0.86 (3 H, d, *J* = 7.0 Hz, H17), 0.70 (3 H, d, *J* = 7.1 Hz, H16). ^13^C NMR (151 MHz, Chloroform-*d*) δ 217.45 (C3), 168.39 (C21), 157.20, 156.97, 156.03, 139.72, 138.98 (C19), 136.94, 132.63, 131.38, 130.33, 121.80, 117.12, 116.27 (C20), 116.12, 114.03, 107.47, 74.56 (C11), 69.75 (C14), 58.18 (C4), 45.43 (C9), 44.77 (C13), 43.87 (C12), 41.78 (C5), 37.59 (C6), 36.77 (C10), 35.99 (C2), 34.49 (C22), 30.37 (C8), 26.82 (C7), 26.56 (C18), 24.81 (C1), 16.74 (C16), 14.88 (C15), 11.50 (C17). HR-MS (ESI): Calcd for C_39_H_43_ClFN_5_O_4_S (M + Cl^−^): 766.2402; Found: 766.2405.

#### 3.2.30. 22-[4-(6-Chloro-1-(2,4-difluorophenyl)-1*H*-pyrazolo[3,4-*d*]pyrimidine-4-yl)amino-Phenylsulfanyl]-22-deoxypleuromutilin (**20c**)

Yellow powder; yield: 51%; melting point: 132–134 °C; ^1^H NMR (600 MHz, Chloroform*-d*) δ 11.04 (1 H, s), 8.37 (1 H, s), 7.61 (2 H, d, *J* = 8.7 Hz), 7.45 (2 H, d, *J* = 8.6 Hz), 7.25–7.17 (1 H, m, 1*H*), 6.99–6.87 (2 H, m), 6.43 (1 H, dd, *J* = 17.4, 11.0 Hz, H19), 5.74 (1 H, d, *J* = 8.5 Hz, H14), 5.33 (1 H, d, *J* = 11.0 Hz, H20), 5.17 (1 H, d, *J* = 18.8 Hz, H20), 3.56 (2 H, d, *J* = 3.7 Hz, H22), 3.36–3.29 (1 H, m, H11), 2.29-1.97 (5H, m, H2, H4, H10, 11-OH), 1.79–1.58 (6 H, m, H1, H6, H7, H8), 1.42 (3 H, s, H15), 1.35–1.25 (2 H, m, H13), 1.12 (4 H, m, H8, H18), 0.86 (3 H, d, *J* = 7.0 Hz, H17), 0.70 (3 H, d, *J* = 7.0 Hz, H16). ^13^C NMR (151 MHz, DMSO*-d_6_*) δ 217.59 (C3), 168.32 (C21), 160.01, 156.79, 155.80, 155.77, 148.48, 148.40, 141.29 (C19), 135.63, 129.45, 129.39, 115.79 (C20), 115.03, 115.00, 111.84, 111.67, 107.51, 104.62, 73.05 (C11), 70.51 (C14), 57.59 (C4), 45.43 (C9), 44.59 (C13), 44.44 (C12), 43.86 (C5), 41.94 (C6), 36.93 (C10), 36.74 (C2), 34.45 (C22), 30.57 (C8), 29.00 (C7), 27.07 (C18), 24.93 (C1), 16.61 (C16), 14.76 (C15), 12.00 (C17). HR-MS (ESI): Calcd for C_39_H_42_ClF_2_N_5_O_4_S (M + Cl^−^): 784.2308; Found: 784.2314

#### 3.2.31. 22-[4-(6-Chloro-1-(2,5-difluorophenyl)-1*H*-pyrazolo[3,4-*d*]pyrimidine-4-yl)amino-Phenylsulfanyl]-22-deoxypleuromutilin (**21c**)

Yellow powder; yield: 77%; melting point: 115–118 °C; ^1^H NMR (600 MHz, Chloroform*-d*) δ 11.14 (1 H, s), 8.75 (1 H, s), 7.68 (2 H, d, *J* = 8.7 Hz), 7.51 (2 H, d, *J* = 8.4 Hz), 7.37–7.28 (1 H, m), 7.00 (1 H, t, *J* = 8.2 Hz), 6.75 (1 H, t, *J* = 8.4 Hz), 6.17 (1 H, dd, *J* = 17.2, 11.7 Hz, H19), 5.63 (1 H, d, *J* = 8.3 Hz, H14), 5.11–5.05 (2 H, m, H20), 3.98–3.82 (2 H, s, H22), 3.58–3.48 (1 H, m, H11), 2.34–2.10 (5 H, m, H2, H4, H10, 11- OH), 1.73–1.26 (11 H, m, H1, H6, H7, H8, H13, H15), 1.09 (4 H, m, H8, H18), 0.90 (3 H, d, *J* = 7.0 Hz, H17), 0.70 (3 H, d, *J* = 7.1 Hz, H16). ^13^C NMR (151 MHz, DMSO*-d_6_*) δ 217.54 (C3), 168.09 (C21), 160.27, 158.68, 157.96, 157.51, 155.83, 146.91, 145.34, 141.12 (C19), 136.58, 136.16, 133.53, 131.51, 130.02, 122.66, 117.11, 115.61 (C20), 107.89, 73.08 (C11), 70.22 (C14), 57.71 (C4), 45.40 (C9), 44.39 (C13), 44.17 (C12), 41.91 (C5), 36.81 (C6, C10), 36.23 (C2), 34.43 (C22), 30.55 (C8), 28.89 (C7), 27.02 (C18), 24.91 (C1), 16.52 (C16), 14.54 (C15), 11.94 (C17). HR-MS (ESI): Calcd for C_39_H_42_ClF_2_N_5_O_4_S (M + Cl^−^): 784.2308; Found: 784.2311.

#### 3.2.32. 22-[4-(6-Chloro-1-(3,4-difluorophenyl)-1*H*-pyrazolo[3,4-*d*]pyrimidine-4-yl)amino-Phenylsulfanyl]-22-deoxypleuromutilin (**22c**)

Yellow powder; yield: 62%; melting point: 109–113 °C; ^1^H NMR (600 MHz, Chloroform*-d*) δ 11.05 (1 H, s), 8.58 (1 H, s), 7.62 (2 H, d, *J* = 8.6 Hz), 7.43 (2 H, d, *J* = 8.6 Hz), 7.12 (1 H, d, *J* = 9.2 Hz), 6.89–6.82 (1 H, m), 6.68 (1 H, d, *J* = 8.5 Hz), 6.42 (1 H, dd, *J* = 17.4, 11.0 Hz, H19), 5.74 (1 H, d, *J* = 8.6 Hz, H14), 5.31 (1 H, d, *J* = 10.8 Hz, H20), 5.17 (1 H, d, *J* = 17.4 Hz, H20), 3.62–3.51 (2 H, m, H22), 3.34 (1 H, d, *J* = 6.3 Hz, H11), 2.35–2.11 (5 H, m, H2, H4, H10, 11-OH), 1.81–1.50 (6 H, m, H1, H6, H7, H8), 1.42 (3 H, s, H15), 1.35 (2 H, m, H13), 1.13 (3 H, s, H18), 1.07 (1 H, m, H8), 0.86 (3 H, d, *J* = 7.0 Hz, H17), 0.70 (3 H, d, *J* = 7.0 Hz, H16). ^13^C NMR (151 MHz, Chloroform-*d*) δ 217.29 (C3), 168.39 (C21), 158.10, 157.19, 156.90, 139.07, 138.90 (C19), 136.79, 134.39, 134.20, 131.58 130.79, 122.01, 118.26, 118.14, 117.30 (C20), 117.15, 106.98, 102.19, 74.62 (C11), 69.73 (C14), 58.18 (C4), 45.46 (C9), 44.79 (C13), 43.89 (C12), 41.75 (C5), 37.66 (C6), 36.79 (C10), 35.99 (C2), 34.49 (C22), 30.42 (C8), 26.84 (C7), 26.37 (C18), 24.83 (C1), 16.79 (C16), 14.88 (C15), 11.50 (C17). HR-MS (ESI): Calcd for C_39_H_42_ClF_2_N_5_O_4_S (M + Cl^−^): 784.2308; Found: 784.2313.

#### 3.2.33. 22-[4-(6-Chloro-1-(4-(yrifluoromethoxy)Phenyl)-1*H*-pyrazolo[3,4-*d*]pyrimidine-4-yl)amino-Phenylsulfanyl]-22-deoxypleuromutilin (**23c**)

Yellow powder; yield: 69%; melting point: 144–147 °C; ^1^H NMR (600 MHz, DMSO*-d_6_*) δ 11.17 (1 H, s), 8.40 (1 H, s), 7.63 (2 H, d, *J* = 8.8 Hz), 7.47 (2 H, d, *J* = 8.8 Hz), 7.32 (2 H, d, *J* = 8.5 Hz), 7.11–7.06 (2 H, d, *J* = 8.5 Hz), 6.10–5.99 (1 H, m, H19), 5.52 (1 H, d, *J* = 8.3 Hz, H14), 4.99–4.95 (2 H, m, H20), 3.91–3.76 (2 H, m, H22),3.34 (1 H, dd, H11), 2.19–1.99 (5 H, m, H2, H4, H10, 11-OH), 1.68-1.36 (5 H, m, H1, H6, H7), 1.34 (3 H, s, H15), 1.31–1.19 (3 H, m, H8, H13), 0.98 (4 H, s, H8, H18), 0.80 (3 H, d, *J* = 7.2 Hz, H17), 0.59 (3 H, d, *J* = 7.0 Hz, H16). ^13^C NMR (151 MHz, DMSO*-d_6_*) δ 217.60 (C3), 168.12 (C21), 157.54, 157.27, 155.38, 143.34, 142.02, 141.17 (C19), 136.16, 133.70, 131.43, 130.20, 123.19, 115.61 (C20), 113.57, 108.19, 73.05 (C11), 70.24 (C14), 57.71 (C4), 45.40 (C13), 44.42 (C12), 44.13, 41.99 (C5), 36.84 (C6), 36.81 (C10), 36.12 (C2), 34.43 (C22), 30.53 (C8), 28.95 (C7), 27.03 (C18), 24.91 (C1), 16.54 (C16), 14.97 (C15), 11.94 (C17). HR-MS (ESI): Calcd for C_40_H_43_ClF_3_N_5_O_5_S (M + Cl^−^): 832.2320; Found: 832.2327.

#### 3.2.34. 22-[4-(6-Chloro-1-(3-chlorophenyl)-1*H*-pyrazolo[3,4-*d*]pyrimidine-4-yl)amino-Phenylsulfanyl]-22-deoxypleuromutilin (**24c**)

Yellow powder; yield: 73%; melting point: 121–124 °C; ^1^H NMR (600 MHz, Chloroform*-d*) δ 11.09 (1 H, s), 8.60 (1 H, s), 7.66 (2 H, d, *J* = 8.6 Hz), 7.40 (2 H, d, *J* = 8.6 Hz), 7.20 (1 H, t, *J* = 8.0 Hz), 7.03 (1 H, s), 6.91 (1 H, d, *J* = 7.9 Hz, 1*H*), 6.79 (1 H, d, *J* = 9.5 Hz), 6.41 (1 H, dd, *J* = 17.4, 11.0 Hz, H19), 5.73 (1 H, d, *J* = 8.5 Hz, H14), 5.29 (1 H, d, *J* = 9.5 Hz, H20), 5.16 (1 H, d, *J* = 17.5 Hz, H20), 3.63–3.52 (2 H, m, H22), 3.33 (1 H, t, *J* = 7.3 Hz, H11), 2.31–2.00 (5 H, m, H2, H4, H10, 11-OH), 1.79–1.43 (6 H, m, H1, H6, H7, H8), 1.42 (3 H, s, H15), 1.34 (2 H, m, H13), 1.13 (3 H, s, H18), 1.07 (1 H, m, H8), 0.86 (3 H, d, *J* = 7.0 Hz, H17), 0.71 (3 H, d, *J* = 7.0 Hz, H16). ^13^C NMR (151 MHz, Chloroform-*d*) δ 217.17 (C3), 168.35 (C21), 158.27, 157.18, 157.05, 144.12, 138.86 (C19), 136.92, 135.67, 134.29, 131.70, 130.77, 130.65, 121.98, 121.56, 117.38 (C20), 112.99, 111.19, 106.93, 74.63 (C11), 69.67 (C14), 58.19 (C4), 45.45 (C9), 44.79 (C13), 43.89 (C12), 41.79 (C5), 37.78 (C6), 36.79 (C10), 35.99 (C2), 34.49 (C22), 30.43 (C8), 26.85 (C7), 26.34 (C18), 24.84 (C1), 16.81 (C16), 14.90 (C15), 11.51 (C17). HR-MS (ESI): Calcd for C_39_H_43_Cl_2_N_5_O_4_S (M + Cl^−^): 784.2077; Found: 784.2079.

#### 3.2.35. 22-[4-(6-Chloro-1-(4-chlorophenyl)-1*H*-pyrazolo[3,4-*d*]pyrimidine-4-yl)amino-Phenylsulfanyl]-22-deoxypleuromutilin (**25c**)

Yellow powder; yield: 65%; melting point: 124–129 °C; ^1^H NMR (600 MHz, Chloroform*-d*) δ 11.11 (1 H, s), 8.33 (1 H, s), 7.61 (2 H, d, *J* = 8.6 Hz), 7.42 (2 H, d, *J* = 8.6 Hz), 7.29 (2 H, d, *J* = 8.7 Hz), 6.95 (2 H, d, *J* = 8.8 Hz), 6.42 (1 H, dd, *J* = 17.4, 11.0 Hz, H19), 5.74 (1 H, d, *J* = 8.5 Hz, H14), 5.31 (1 H, d, *J* = 12.3 Hz, H20), 5.17 (1 H, d, *J* = 17.4 Hz, H20), 3.62–3.54 (2 H, m, H22), 3.33 (1 H, d, *J* = 6.3 Hz, H11), 2.34–2.08 (5 H, m, H2, H4, H10, 11-OH), 1.81–1.51 (6 H, m, H1, H6, H7, H8), 1.42 (3 H, s, H15), 1.39–1.30 (2 H, m, H13), 1.12 (3 H, d, *J* = 7.0 Hz, H18), 1.08 (1 H, m, H8), 0.86 (3 H, d, *J* = 7.0 Hz, H17), 0.70 (3 H, d, *J* = 7.1 Hz, H16). ^13^C NMR (151 MHz, Chloroform-*d*) δ 217.16 (C3), 168.35 (C21), 158.03, 157.21, 156.87, 141.66 (C19), 138.91, 136.87, 133.85, 131.60, 130.73, 129.70, 126.52, 122.09, 117.33 (C20), 114.13, 107.02, 74.62 (C11), 69.69 (C14), 58.18 (C4), 45.45 (C9), 44.79 (C13), 43.90 (C12), 41.78 (C5), 37.71 (C6), 36.77 (C10), 35.99 (C2), 34.48 (C22), 30.41 (C8), 26.85 (C7), 26.37 (C18), 24.83 (C1), 16.80 (C16), 14.89 (C15), 11.50 (C17). HR-MS (ESI): Calcd for C_39_H_43_Cl_2_N_5_O_4_S (M + Cl^−^): 784.2077; Found: 784.2076.

#### 3.2.36. 22-[4-(6-Chloro-1-(3,5-dichlorophenyl)-1*H*-pyrazolo[3,4-*d*]pyrimidine-4-yl)amino-Phenylsulfanyl]-22-deoxypleuromutilin (**26c**)

Yellow powder; yield: 73%; melting point: 137–139 °C; ^1^H NMR (600 MHz, Chloroform*-d*) δ 11.01 (1 H, s), 8.29 (1 H, s), 7.71 (2 H, d, *J* = 8.7 Hz), 7.46 (2 H, d, *J* = 8.7 Hz), 6.97 (1 H, t, *J* = 1.7 Hz), 6.93 (2 H, d, *J* = 1.7 Hz), 6.42 (1 H, dd, *J* = 17.4, 11.0 Hz, H19), 5.74 (1 H d, *J* = 8.5 Hz, H14), 5.32 (1 H, d, *J* = 11.0 Hz, H20), 5.17 (1 H, d, *J* = 16.1 Hz, H20), 3.61–3.53 (2 H, m, H22), 3.38–3.28 (1 H, m, H11), 2.31–1.99 (5 H, m, H2, H4, H10, 11-OH), 1.73–1.42 (6 H, m, H1, H6, H7, H8), 1.41 (3 H, s, H15), 1.36 (2 H, m, H13), 1.12 (4 H, s, H8, H18), 0.86 (3 H, d, *J* = 7.0 Hz, H17), 0.70 (3 H, d, *J* = 7.0 Hz, H16). ^13^C NMR (151 MHz, Chloroform-*d*) δ 217.10 (C3), 168.28 (C21), 158.82, 157.54, 157.22, 144.66, 138.86 (C19), 136.75, 136.23, 135.55, 131.72, 130.90, 121.93, 121.38, 117.37 (C20), 111.40, 106.58, 74.63 (C11), 69.65 (C14), 58.18 (C4), 45.45 (C9), 44.80 (C13), 43.89 (C12), 41.78 (C5), 37.77 (C6), 36.78 (C10), 35.99 (C2), 34.47 (C22), 30.43 (C8), 26.85 (C7), 26.32 (C18), 24.84 (C1), 16.80 (C16), 14.88 (C15), 11.50 (C17). HR-MS (ESI): Calcd for C_39_H_42_Cl_3_N_5_O_4_S (M + Cl^−^): 818.1688; Found: 818.1698.

#### 3.2.37. 22-[4-(6-Chloro-1-(3-nitrophenyl)-1*H*-pyrazolo[3,4-*d*]pyrimidine-4-yl)amino-Phenylsulfanyl]-22-deoxypleuromutilin (**27c**)

Yellow powder; yield: 74%; melting point: 137–139 °C; ^1^H NMR (600 MHz, Chloroform*-d*) δ 10.90 (1 H, s), 8.79 (1 H, s), 7.87 (1 H, s), 7.75 (3 H, d, *J* = 8.5 Hz), 7.43 (3 H, d, *J* = 8.5 Hz), 7.16 (1 H, d, *J* = 9.3 Hz), 6.41 (1 H, dd, *J* = 17.4, 11.0 Hz, H19), 5.73 (1 H, d, *J* = 8.5 Hz, H14), 5.27 (1 H, d, *J* = 11.4 Hz, H20), 5.17 (1 H, d, *J* = 17.4 Hz, H20), 3.62-3.52 (2 H, m, H22), 3.35 (1 H, s, H11), 2.31–2.03 (5 H, m, H2, H4, H10, 11-OH), 1.80–1.46 (6 H, m, H1, H6, H7, H8), 1.43 (3 H, s, H15), 1.40–1.26 (2H, m, H13), 1.13 (4 H, m, H8, H18), 0.86 (3 H, d, *J* = 7.0 Hz, H17), 0.73 (3 H, d, *J* = 7.0 Hz, H16). ^13^C NMR (151 MHz, DMSO*-d_6_*) δ 216.83 (C3), 168.12 (C21), 158.32, 156.96, 156.64, 148.96, 143.60, 138.43 (C19), 136.25, 135.29, 131.21, 130.40, 130.02, 121.52, 118.17, 116.92 (C20), 115.42, 106.19, 74.19 (C11), 69.42 (C11), 57.73 (C4), 45.03 (C9), 44.32 (C13), 43.51 (C12), 41.38 (C5), 37.21 (C6), 36.37 (C10), 35.55 (C2), 34.08 (C22), 30.00 (C8), 26.44 (C7), 25.90 (C18), 24.40 (C1), 16.45 (C16), 14.46 (C15), 11.06 (C17). HR-MS (ESI): Calcd for C_39_H_43_ClN_6_O_6_S (M + Cl^−^): 793.2347; Found: 793.2353.

#### 3.2.38. 22-[4-(6-Chloro-1-(naphthalen-2-yl)-1*H*-pyrazolo[3,4-*d*]pyrimidine-4-yl-yl)amino-Phenylsulfanyl]-22-deoxypleuromutilin (**28c**)

Yellow powder; yield: 63%; melting point: 115–118 °C; ^1^H NMR (600 MHz, Chloroform*-d*) δ 11.32 (1 H, s), 8.33 (1 H, s), 7.79–7.70 (4 H, m, 4H), 7.64 (1 H, d, *J* = 8.2 Hz), 7.45 (3 H, d, *J* = 8.0 Hz), 7.34 (2 H, s), 7.15 (1 H, d, *J* = 8.6 Hz), 6.42 (1 H, dd, *J* = 17.4, 11.0 Hz, H19), 5.74 (1 H, d, *J* = 8.4 Hz, H14), 5.28(1 H, m, H20), 5.14 (1 H, d, *J* = 17.4 Hz, H20), 3.63–3.50 (2 H, m, H22), 3.31 (1 H, d, *J* = 5.3 Hz, H11), 2.31–2.02 (5 H, m, H2, H4, H10, 11-OH), 1.77–1.48 (6 H, m, H1, H6, H7, H8), 1.43 (3 H, s, H15), 1.30 (2 H, m, H13), 1.14 (1 H, m, H8), 1.10 (3 H, s, H18), 0.84 (3 H, d, *J* = 6.9 Hz, H17), 0.71 (3 H, d, *J* = 7.1 Hz, H16). ^13^C NMR (151 MHz, Chloroform-*d*) δ 217.18 (C3), 168.33 (C21), 157.74, 157.10, 156.58, 140.52, 138.84 (C19), 137.14, 134.38, 134.02, 133.33, 131.72, 130.43, 129.79, 129.45, 127.92, 127.21, 126.39, 124.03, 121.98, 117.36 (C20), 114.96, 107.51, 74.59 (C11), 69.65 (C14), 58.17 (C4), 45.43 (C9), 44.77 (C13), 43.87 (C12), 41.77 (C5), 37.80 (C6), 36.77 (C10), 35.97 (C2), 34.46 (C22), 30.40 (C8), 26.84 (C7), 26.34 (C18), 24.81 (C1), 16.80 (C16), 14.90 (C15), 11.49 (C17). HR-MS (ESI): Calcd for C_43_H_46_ClN_5_O_4_S (M + Cl^−^): 798.2653; Found: 798.2651.

### 3.3. In Vitro Efficacy of Pleuromutilin Derivatives

#### 3.3.1. Minimal Inhibitory Concentration (MIC) Testing and Minimum Bactericidal 

##### Concentration (MBC) Testing

The MIC and MBC values of the synthesized novel pleuromutilin derivatives against MRSA ATCC 43300, *S. aureus* ATCC 29213, *S. aureus* 144 and *S. aureus* AD3 were measured. Tiamulin was used as a positive control drug. The MIC and MBC values were manipulated by broth dilution according to the Clinical and Laboratory Standards (CLSI). Each compound to be measured was dissolved in a solution of 95% deionized water, 2.5% dimethyl sulfoxide (DMSO) and 2.5% Tween-80 with a concentration of 1280 µg/mL. Subsequently, serial twofold dilutions were made in a concentration range from 64 µg/mL to 0.0625 µg/mL. The working bacterial suspension was inoculated into each well, which provided the final inoculum density of 5 × 10^5^ CFU/mL. Three parallel experiments were performed for each compound. The MIC value was recorded as the lowest inhibitory concentration of the sample on the visible growth of the tested bacteria after 24 h of incubation at 37 °C.

After obtaining the MIC results, the 96-well plates were incubated at 37 °C for 24 h. Overall, 25 µL of the bacterial solution from the wells with no obvious bacterial growth was inoculated on MH agar plates [45]. Then, the inoculated MH agar plates were further cultured at 37 °C for 24 h. MBC was defined as the minimum concentration at which bacterial growth was not observed (99.9% of the bacteria were killed).

#### 3.3.2. Constant Concentration Time–Kill Curves

Time–kill curve assays were established with the MIC value to investigate the antibacterial effect of constant drug concentration on MRSA. The antibiotic concentrations were equal to 1 × MIC, 2 × MIC, 4 × MIC, 8 × MIC, 16 × MIC and 32 × MIC as described. MRSA ATCC 43300 was incubated in MH broth at 37 °C for 4.5 h and diluted to 1 × 10^6^ CFU/mL in MH broth. The saline control group was subjected to the same schedule but 0.9% saline water was added instead of the test compound. All the samples were incubated in an oscillating thermostatic at 37 °C, and then 100 μL of the mixture was extracted to 900 μL of sterile saline (0.9%) at 0, 3, 6, 9, 12 and 24 h, respectively. Samples were serially diluted 10-fold in sterile saline (0.9%) and inoculated onto agar plates. The colonies were counted after incubation at 37 °C for 24 h. Three independent experiments were performed according to our previous work [34]. Time–kill curves were constructed by plotting the log_10_ CFU per millilitre versus time, and the change in bacterial concentration was determined.

#### 3.3.3. Determination of the Post-Antibiotic Effect (PAE)

PAE of compounds **12c** and **22c** on MRSA ATCC 43300 was determined using MH broth according to our previous work [46]. The final concentration of MRSA was 1 × 10^6^ CFU/mL by dilution with MH broth. Compounds **12c**, **22c** and tiamulin were supplemented in the suspension at a final concentration of 2 × MIC and 4 × MIC, respectively. The negative control group contained untreated MRSA bacterial cells. The test tubes were incubated with a 37 °C constant temperature vibration incubator for 1 and 2 h. After incubation, the test compound was removed from the sample by diluting 1000-fold with the preheated MH broth. Next, 100 μL of suspension from each culture was 10-fold diluted in sterile saline and inoculated on MH agar plates at 0, 2, 4, 6 and 8 h after inoculation. The number of colonies was calculated after incubation at 37 °C for 24 h. The experiments were performed in triplicate.

### 3.4. Cytotoxicity Assay

Cytotoxicity of the test compounds was assessed using the conventional MTT method [46]. RAW 264.7 murine macrophage cells, Caco-2 cells and 16-HBE were used in this experiment. The cells were seeded into 96-well plates at a density of 1.0 × 10^5^ cells per well. After 4 h at 37 °C, the cells were treated with compounds **12c** and **22c** at various concentrations and incubated at 37 °C for 16 h. Next, MTT (0.5 mg/mL in PBS, 100 μL/well) was added sequentially to each well in a humidified atmosphere of 95% air and 5% CO_2_ incubator and the incubation continued for an additional four hours. After incubation, the medium was peeled off and DMSO (150 μL/well) was added to dissolve the cells, and then the incubation was continued for 30 min. At the end of this period, the absorbance was measured at 490 nm by a microplate spectrophotometer (BIO-TEK Instruments, Winooski, VT, USA).

### 3.5. Neutropenic Murine Thigh Infection Model

Model experiments of neutropenic mouse thigh infection were manipulated as described in the literature [32]. Female, six-week-old, specific-pathogen-free mice weighing approximately 23~26 g were used throughout the study. Mice were rendered neutropenic by injection of cyclophosphamide (Mead Johnson Pharmaceuticals, Evansville, IN) on day 4 (150 mg/kg) and day 1 (100 mg/kg) before the experiment. The neutrophil numbers in mice blood should be <0.1 × 10^9^/L [34]. In total, 0.1 mL of MH broth of MRSA (10^7^ CFU/mL) was inoculated into each thigh of the neutropenic mice. Mice were divided into 4 groups (6 mice per group), including control (sterile saline), compounds **12c**, **22c** and tiamulin groups. After 3 h post-infection, the corresponding drugs were injected into each mouse’s thigh. Mice were euthanized following intravenous injection for 24 h. Thigh tissue of each mouse was removed, collected, weighed and homogenized in 3 mL of ice sterile saline. Six ten-fold serial dilutions were performed and 25 μL of the bacterial solution from each tube was plated on MH agar plates. The resulting colonies were counted after 24 h incubation at 37 °C. The protocol for this study was reviewed and approved by the Institutional Animal Care and Use Committee of the South China Agricultural University.

### 3.6. Molecular Modeling

Docking studies were carried out based on the binding mode of the *Staphylococcus aureus* 50S ribosome with tiamulin (PDB ID code: 1XBP). The binding pattern of compound **22c** to *S. aureus* 50S ribosome was investigated. All residues within 40 Å around tiamulin in 1XBP were built as a peptidyl transferase centre (PTC) model. The model was then refined using a standard energy minimization protocol. The test compound was prepared by Avogadro 1.1.1, with a 5000-step Steepest Descent as well as a 1000-step Conjugate Gradients geometry optimization using MMFF94 force field. Docking experiments were performed using AutoDock, Vina and Pymol [47].

## 4. Conclusions

Two series of pleuromutilin derivatives containing 6-chloro-4-amino-1-R-1*H*-pyrazolo[3,4-*d*]pyrimidine structures were synthesized and evaluated as inhibitors against MRSA. The present study revealed the synthesis procedure, antibacterial activity and cytotoxicity of the designed compounds. Compound **22c** in Figure 2 displayed better antibacterial activity against MRSA than tiamulin. The resulting time–kill curve experiments indicated that compound **22c** was time-dependent rather than dose-dependent and manifested a more rapid bactericidal kinetic effect than tiamulin. Compound **22c** performed longer PAE than tiamulin, indicating a longer administration interval than tiamulin. The results of the cytotoxicity assay revealed that compound **22c** exhibited no significant inhibitory effect on RAW 264.7 cells, Caco-2 cells and 16-HBE cells at high doses. Meanwhile, compound **22c** exhibited more potent in vivo bactericidal effects than tiamulin in the neutropenic murine thigh infection model studies. Moreover, the molecular docking studies indicated that four hydrogen bonds played important roles in the binding of compound **22c** to 50S ribosomes. This study indicated that compound **22c** was worthy of further development as a potential drug against MRSA infection.

## Data Availability

Data available on request due to restrictions privacy. The data presented in this study are available on request from the corresponding author.

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
