# Peer review of "Design, Synthesis and Biological Evaluation of Novel Pleuromutilin Derivatives Containing 6-Chloro-1-R-1H-pyrazolo[3,4-d]pyrimidine-4-amino Side Chain"

_molecules, 2023, doi:10.3390/molecules28093975_

Round 1
Reviewer 1 Report
The synthesis of new pleuromutilin derivatives has been described. Some derivatives have been shown to have antibacterial activity.
The manuscript is well-structured, interesting and contains new data. This work should be improved according to the following comments:
1) It is necessary to correct typos that occur in the text.
2) In Abstract decipher the abbreviation «MRSA» and «PAE».
3) L.21: Please, check if “superior” is appropriate for comparison of antibacterial activity between the compounds with MIC = 0.25 μg/mL and tiamulin with MIC = 0.5 μg/mL.
4) Figure 1. Please, provide a picture of Pleuromutilin in higher quality and with clearer numbering. In the title compounds 7 and 8 are missing.
5) Scheme 1 and further in the text, is it necessary to give the yields with 2 decimals?
6) In Scheme 2: (v) “DCM” should be replaced by “EtOAc” according to “4.2.3. 22-(4-Amino-phenylsulfanyl)deoxypleuromutilin (32)”
7) L. 169: “3.78” should be replaced by “-3.78”.
8) Figure 5. Please provide p values.
9) L. 302: Please specify the “alcohol”.
10) L. 327: Check the solvent mixture for column chromatography “(dichloromethane: methanol = 1:200)”.
Author Response
Responses to Referee #1:
Reviewers' comments:
Reviewer #1: Tang et. al. reported two series of pleuromutilin derivatives, which are derived from 6-chloro-4-amino-1-R-1H-pyrazolo[3,4-d]pyrimidine or 4-(6-chloro-1-R-1H-pyrazolo[3,4-d] pyrimidine-4-ly)amino-phenylthiol with pleuromutilin. And in vitro antibacterial activities of these semisynthetic derivatives were evaluated. Among them, three compounds 12c, 19c and 22c manifested superior antibacterial ability against MRSA than tiamulin. In addition, compound 22c exhibited longer PAE than tiamulin, and showed no significant inhibition on the cell viability. This research is a useful addition to the medicinal community. This research is considered to be published after major revisions:
1.The SAR should be further discussed and clarified. For example, the antibacterial activity of 6-chloro-4-amino-1-R-1H-pyrazolo[3,4-d]pyrimidine, 4-aminothiophenol and parent compound pleuromutilin should be included in manuscript.
Response: Many thanks for the reviewer’s comments. The MIC of 6-chloro-4-amino-1-R-1H-pyrazolo[3,4-d]pyrimidine, 4-aminothiophenol was ≥ 64, and the result has been added in the supporting information. We added SAR of these derivatives in the revised manuscript (page 10, line 158-~165)
Compound 22b seems to show less biological activity than compound 22c. How to explain this SAR? What is the essential group to affect the activity in these compounds?
Response: Many thanks for the reviewer’s comments. The activity of compound 22c was better than compound 22b, this may be associated with the introduction of 4-aminothiophenol. This suggested that the formation of thioether bond at the C22 position could enhance the antibacterial activity of pleuromutilin, which was consistent with the literature reports and our previous work. Considering pleuromutilin derivatives without thioether at the C22 position displayed weak inhibitory effect on CYP3A4(Molecules 2022, 27, 931(doi:10.3390/molecules27030931)), we focused on developing pleuromutilin derivatives with amine group in this manuscript.
Suggest the “NH” group in the compounds 12b ~ 28b is replaced by “S” atom and then evaluate the antibacterial activity.
Response: Many thanks for the reviewer’s comments. Introducing “S” atom into C22 side chain of pleuromutilin is a common and effective idea to modifying pleuromutilin. However, literatures have also shown that these target pleuromutilin derivatives containing “S” atom can reduce the inhibiting effect on CYP450 enzyme. According to the previous work in our team, part of pleuromutilin derivatives (containing amino side chains) can exhibit better antibacterial activity (MIC=0.06~0.5 μg/mL) and weak inhibition of CYP450 enzyme compared with tiamulin (doi: 10.3390/molecules27030931). Therefore, the modification strategy which introducing amino substitution on C22 is considered as effective. Based on this, we try to introduce NH group to pleuromutilin and investigate their antibacterial activity in vivo and in vitro. Unfortunately, we don’t obtain target derivative with excellent antibacterial activity as expected. We will summarize and analyze the reasons of the poor antibacterial activity combining with SAR, antibacterial activity and reviewer’s suggestion. And we have added more analysis and discussion in the revised manuscript (page 10, line 157~163). Many thanks to the reviewer’s suggestion to improve our manuscript. We will try to introduce “S” atom to these derivatives and our other works in future according to the reviewer's suggestion!
- Some reaction conditions described in Scheme 1 and 2 are not consistent with those in experimental procedures. For example, the sentence “Compound 29 (10 g, 18.8 mmol) and sodium iodide (3.1 g, 20.7 mmol) was added in ethyl acetate (100 mL).” But, in Scheme 2, the solvent is DCM not ethyl acetate.
Response: Many thanks for the reviewer’s comments. We have revised the reaction conditions of scheme 1 and 2 (page 4, line114) and method section in the revised manuscript.
The sentence “20% aqueous NaOH (10 mL) with p-aminothiophenol (3.5 g, 27.9 mmol) was added dropwise……” But, in Scheme 2, the corresponding condition is “K2CO3, DCM” not “20% aqueous NaOH in ethyl acetate”. Anyway, there are such many errors in the manuscript.
Response: Many thanks for the reviewer’s comments. We feel so sorry about these errors. We have delated “K2CO3” and revised the reaction conditions as “20% aqueous NaOH, DCM” in scheme 2 (page 4, line 113) and the sentence in method section (page 18, line 346~347) in the revised manuscript. And We have carefully checked full text ensured they were consistent in the revised manuscript.
- In experiment section, the eluent solvent in column chromatography purification is dichloromethane: methanol = 1:200. It seems to be mistaken.
Response: Many thanks the reviewer’s comments. The eluent solvent is “dichloromethane: methanol = 200:1.” We have corrected this mistake in the revised manuscript (page 18, lines 343,350,356)
- Suggest to include the detailed procedure for the recrystallization of compound 29.
Response: Many thanks the reviewer’s comments. Following the suggestion of the reviewer, the detailed procedure for the recrystallization of compound 29 was added in section 4.2.2 (page 17, lines 327~331) in the revised manuscript.
- Suggest to include the detailed Vilsmeier-Haack reaction procedure for the preparation of compound 11.
Response: Many thanks the reviewer’s comments. The detailed Vilsmeier-Haack reaction procedure for the preparation of compound 11 was added in section 4.2.1 in the revised manuscript (page 17, lines 310~315).
- Some reagents are missed in the preparation of compound 31 from 30.
Response: Many thanks the reviewer’s comments. Missed reagents H2O and triphenylphosphine had been added in the revised manuscript (page 18, lines 338~339).
- Mp was provided for each solid compound.
Response: Many thanks the reviewer’s comments. The melting point of all compounds have been shown as following and added in section 4.2.5-4.2.38 in the revised manuscript.
|
Compound No |
melting point |
Compound No |
melting point |
|
12b |
97-99°C |
12c |
95-97°C |
|
13b |
93-95°C |
13c |
101-105°C |
|
14b |
99-103°C |
14c |
103-105°C |
|
15b |
95-100°C |
15c |
92-95°C |
|
16b |
101-103°C |
16c |
97-99°C |
|
17b |
99-104°C |
17c |
97-102°C |
|
18b |
107-110°C |
18c |
123-127°C |
|
19b |
112-115°C |
19c |
141-144°C |
|
20b |
123-125°C |
20c |
132-134°C |
|
21b |
121-125°C |
21c |
115-118°C |
|
22b |
130-132°C |
22c |
109-113°C |
|
23b |
132-125°C |
23c |
144-147°C |
|
24b |
117-119°C |
24c |
121-124°C |
|
25b |
107-111°C |
25c |
124.129°C |
|
26b |
151-156°C |
26c |
137-139°C |
|
27b |
131-135°C |
27c |
137.139°C |
|
28b |
123-125°C |
28c |
115-118°C |
- The actronym “NET3” should be revised into “Et3N”.
Response: Many thanks for the reviewer’s comments. We have carefully checked all NET3 errors and revised as “Et3N” (page 4, Line107, 109, 113 and 115) in the revised manuscript.
- English language should be improved and edited throughout the whole MS.
Response: Many thanks for the reviewer’s comments. We have carefully checked and revised the English grammar and format in the revised manuscript.

Reviewer 2 Report
Tang et. al. reported two series of pleuromutilin derivatives, which are derived from 6-chloro-4-amino-1-R-1H-pyrazolo[3,4-d]pyrimidine or 4-(6-chloro-1-R-1H-pyrazolo[3,4-d] pyrimidine-4-ly)amino-phenylthiol with pleuromutilin. And in vitro antibacterial activities of these semisynthetic derivatives were evaluated. Among them, three compounds 12c, 19c and 22c manifested superior antibacterial ability against MRSA than tiamulin. In addition, compound 22c exhibited longer PAE than tiamulin, and showed no significant inhibition on the cell viability. This research is a useful addition to the medicinal community. This research is considered to be published after major revisions:
1. The SAR should be further discussed and clarified. For example, the antibacterial activity of 6-chloro-4-amino-1-R-1H-pyrazolo[3,4-d]pyrimidine, 4-aminothiophenol and parent compound pleuromutilin should be included in manuscript. Compound 22b seems to show less biological activity than compound 22c. How to explain this SAR? What is the essential group to affect the activity in these compounds? Suggest the “NH” group in the compounds 12b ~ 28b is replaced by “S” atom and then evaluate the antibacterial activity.
2. Some reaction conditions described in Scheme 1 and 2 are not consistent with those in experimental procedures. For example, the sentence “Compound 29 (10 g, 18.8 mmol) and sodium iodide (3.1 g, 20.7 mmol) was added in ethyl acetate (100 mL).” But, in Scheme 2, the solvent is DCM not ethyl acetate. The sentence “20% aqueous NaOH (10 mL) with p-aminothiophenol (3.5 g, 27.9 mmol) was added dropwise……” But, in Scheme 2, the corresponding condition is “K2CO3, DCM” not “20% aqueous NaOH in ethyl acetate”. Anyway, there are such many errors in the manuscript.
3. In experiment section, the eluent solvent in column chromatography purification is dichloromethane: methanol = 1:200. It seems to be mistaken.
4. Suggest to include the detailed procedure for the recrystallization of compound 29.
5. Suggest to include the detailed Vilsmeier-Haack reaction procedure for the preparation of compound 11.
6. Some reagents are missed in the preparation of compound 31 from 30.
7. Mp was provided for each solid compound.
8. The actronym “NET3” should be revised into “ Et3N”.
9. English language should be improved and edited throughout the whole MS.
Author Response
Responses to Referee #2:
Reviewers' comments:
The synthesis of new pleuromutilin derivatives has been described. Some derivatives have been shown to have antibacterial activity.
The manuscript is well-structured, interesting and contains new data. This work should be improved according to the following comments:
1) It is necessary to correct typos that occur in the text.
Response: Many thanks for the reviewer’s comments. We have carefully checked the English grammar, typos and format in the revised manuscript.
2) In Abstract decipher the abbreviation «MRSA» and «PAE».?
Response: Many thanks the reviewer’s comments. We have deleted “Methicillin-resistant Staphylococcus aureus and post-antibiotic effect” in Abstract section, and used abbreviation in the revised manuscript (page 1, lines 20, 25).
3) L.21: Please, check if “superior” is appropriate for comparison of antibacterial activity between the compounds with MIC = 0.25 μg/mL and tiamulin with MIC = 0.5 μg/mL.
Response: Many thanks the reviewer’s comments. We have corrected this inappropriate description in original manuscript, and this sentence changed as “Compounds 12c, 19c and 22c (MIC=0.25 μg/mL) manifested good in vitro antibacterial ability against MRSA which was similar to that of tiamulin (MIC=0.5 μg/mL)” in the revised manuscript (page 1, line 21~23).
4) Figure 1. Please, provide a picture of Pleuromutilin in higher quality and with clearer numbering. In the title compounds 7 and 8 are missing.
Response: Many thanks for the reviewer’s comments. The number and structure of the compounds 7 and 8 have been newly added, and replaced by higher quality and with clearer numbering in the revised manuscript (page 2, lines 64).
5) Scheme 1 and further in the text, is it necessary to give the yields with 2 decimals?
Response: Many thanks for the reviewer’s comments. The results were calculated based on the actual reaction.
6) In Scheme 2: (v) “DCM” should be replaced by “EtOAc” according to “4.2.3. 22-(4-Amino-phenylsulfanyl)deoxypleuromutilin (32)”
Response: Many thanks for the reviewer’s comments. The solvent has been corrected in Scheme 2 and 4.2.3 section in the revised manuscript (page 18, 347~351).
7) L. 169: “3.78” should be replaced by “-3.78”.
Response: Many thanks for the reviewer’s comments. The “3.78” has been replaced by “-3.78” (page 18, line 177) in the revised manuscript.
8) Figure 5. Please provide p values.
Response: Many thanks for the reviewer’s comments. The p values of Figure 5 was shown as following table. We apologize for the error in Figure 5 (the mistake of significance analysis between “Tiamulin group” and “compound 12c group”). We replaced original error figure by the correct picture in the revised manuscript (page 15, line 246).
|
Group |
P value |
P value summary |
|
Growth Control vs compound 12c |
<0.0001 |
**** |
|
Growth Control vs compound 22c |
<0.0001 |
**** |
|
Tiamulin vs compound 12c |
0.0009 |
*** |
|
Tiamulin vs compound 22c |
0.0001 |
*** |
9) L. 302: Please specify the “alcohol”.
Response: Many thanks the reviewer’s comments. We have deleted the inappropriate description. We replaced "alcohol" by “EtOH” in the revised manuscript (page 17, lines 306, 318, 321)
10) L. 327: Check the solvent mixture for column chromatography “(dichloromethane: methanol = 1:200)”.
Response: Many thanks the reviewer’s comments. We have revised this mistake in the revised manuscript (page 18, lines 343, 350, 356)
Reviewer 3 Report
The manuscript of J. Wang et al. present design, synthesis and biological evaluation of two series pleuromutilin derivatives containing 6-chloro-1-R-1H-pyrazolo[3,4-d]pyrimidine-4-amino side chain. The manuscript is sufficiently novel and in the scope of the journal to warrant publication. However, according to my modest opinion, the manuscript needs revision.
Please find below some of the comments.
According to the IUPAC Nomenclature the names of synthesized compounds should be corrected. For example for compound 12b it should be 22-[(6-chloro-1-(3-methylphenyl)-1H-pyrazolo[3,4-d]pyrimidine-4-yl)amino]-22-deoxypleuromutilin. Please correct the other names as well.
The authors say ²There were about 120,000 bloodstream infections caused by MRSA in the USA in 2017, and nearly 20,000 of them lost their lives [4,5]”. However, the cited literature is from 2012. Also, they say ²Pleuromutilin (1, Figure 1), containing a tricyclic core of five-, six-, and eight-membered rings, was first isolated from two basidiomycete species Pleurotus mutilus and Pleurotus passeckerianus in 1951 [7] ”. However, reference 7 does not mention pleuromutilin.
I propose to delete in scheme 2 the preparation of compounds 12a-28a and 29 because it is given in scheme 1.
The authors say that they used triethylsilane as an internal standard. Maybe it's a mistake and it should be tetramethylsilane.
Some data in the experimental part are different from those in the legend of Scheme 1 and 2. It is necessary to match.
Line 65, instead of (6), it should be (8).
Lines 106, 108 and 114, instead of NET3, it should be NEt3.
Line 298, instead of PH, it should be pH.
Line 316 is missing triphenylphosphine .
Lines 455 and 468, instead of C39H42ClF2N5O4S, it should be C33H38ClF2N5O4.
Line 546, instead of C33H39ClN6O6S, it should be C33H39ClN6O6.
Check it out for HRMS of compound 17c, line 645. Look supplementary data.
Author Response
Responses to Referee #3:
Reviewers' comments:
The manuscript of J. Wang et al. present design, synthesis and biological evaluation of two series pleuromutilin derivatives containing 6-chloro-1-R-1H-pyrazolo[3,4-d]pyrimidine-4-amino side chain. The manuscript is sufficiently novel and in the scope of the journal to warrant publication. However, according to my modest opinion, the manuscript needs revision.
Please find below some of the comments.
- According to the IUPAC Nomenclature the names of synthesized compounds should be corrected. For example for compound 12b it should be 22-[(6-chloro-1-(3-methylphenyl)-1H-pyrazolo[3,4-d]pyrimidine-4-yl)amino]-22-deoxypleuromutilin. Please correct the other names as well.
Response: Thanks the reviewer’s comments. We have corrected the names of all compounds according to the IUPAC Nomenclature in the revised manuscript.
- The authors say ²There were about 120,000 bloodstream infections caused by MRSA in the USA in 2017, and nearly 20,000 of them lost their lives [4,5]”. However, the cited literature is from 2012.
Response: Thanks the reviewer’s comments. We have corrected and cited new reference (Morb Mortal Wkly Rep, 2019, 68, 214~219, doi:10.15585/mmwr.mm6809e1) in the revised manuscript (page 1, line 42 and page 30,952~955).
Also, they say ²Pleuromutilin (1, Figure 1), containing a tricyclic core of five-, six-, and eight-membered rings, was first isolated from two basidiomycete species Pleurotus mutilus and Pleurotus passeckerianus in 1951 [7]”. However, reference 7 does not mention pleuromutilin.
Response: Thanks the reviewer’s comments. We have corrected and cited new reference (Proc Natl Acad Sci USA 1951, Sep,3,9,570-574. doi: 10.1073/pnas.37.9.570.) in the revised manuscript (page 2, line 48 and page 30, 958~960).
- I propose to delete in scheme 2 the preparation of compounds 12a-28a and 29 because it is given in scheme 1.
Response: Many thanks for the reviewer’s comments. We deleted the synthetic method of compounds 12a-28a and 29 and changed new figure of scheme 2 in the revised manuscript (page 4, line 111). And new scheme 2 was shown as following.
- The authors say that they used triethylsilane as an internal standard. Maybe it's a mistake and it should be tetramethylsilane.
Response: Many thanks for the reviewer’s comments. We have corrected this error, and replaced “triethylsilane (in original manuscript)” by “tetramethylsilane” in the revised manuscript (page 17, line 297)
- Some data in the experimental part are different from those in the legend of Scheme 1 and 2. It is necessary to match.
Response: Many thanks for the reviewer’s comments. We feel so sorry about these errors in Scheme 1 and 2 and full text. And we have carefully checked our original manuscript, then tagged and corrected these errors in revised manuscript.
- Line 65, instead of (6), it should be (8).
Response: Many thanks for the reviewer’s comments. The number and structure of the compounds have been corrected in revised manuscript (page 2, line 66).
- Lines 106, 108 and 114, instead of NET3, it should be NEt3.
Response: Many thanks for the reviewer’s comments. We have carefully checked all NET3 errors and revised as “Et3N” (page 4, Line107, 109, 113 and 115 in the revised manuscript.
- Line 298, instead of PH, it should be pH.
Response: Many thanks the reviewer’s comments. We have changed “PH” as “pH” in in the revised manuscript (page 7, line 310).
- Line 316 is missing triphenylphosphine.
Many thanks the reviewer’s comments. Missed reagents H2O and triphenylphosphine had been added in the revised manuscript (page 18, lines 338~339).
- Lines 455 and 468, instead of C39H42ClF2N5O4S, it should be C33H38ClF2N5O4.
Response: Many thanks the reviewer’s comments. “C39H42ClF2N5O4S” have been changed into “C33H38ClF2N5O4” in the revised manuscript (page 20, line 481).
- Line 546, instead of C33H39ClN6O6S, it should be C33H39ClN6O6.
Response: Many thanks the reviewer’s comments. We have corrected this error. “C33H39ClN6O6”has been added in the revised manuscript (page 22, line 577).
- Check it out for HRMS of compound 17c, line 645. Look supplementary data.
Response: Many thanks the reviewer’s comments. According to the mass spectra of compound 17c have been updated as “778.2604” in the revised manuscript (page 24, line 678).
Round 2
Reviewer 2 Report
The author has answered all of reviewer's comments well and corrected the corresponding sections in the revision. So I recommend to be accepted after minor revision.
Comments: Pls checck if the description of HRMS C34H42ClN5O4 (M+Cl- ) is correct.
Author Response
Response: Many thanks for the reviewer’s comments. We carefully examined this description. The molecular weight of this compound as well as the HRMS results has been checked.